# Exercise in Diabetic Cardiomyopathy: Its Protective Effects and Molecular Mechanism

**DOI:** 10.3390/ijms26041465

**Published:** 2025-02-10

**Authors:** Humin Chen, Liang Guo

**Affiliations:** 1School of Exercise and Health and Collaborative Innovation Center for Sports and Public Health, Shanghai University of Sport, Shanghai 200438, China; 2321517010@sus.edu.cn; 2Shanghai Frontiers Science Research Base of Exercise and Metabolic Health, Shanghai University of Sport, Shanghai 200438, China; 3Key Laboratory of Exercise and Health Sciences of the Ministry of Education, Shanghai University of Sport, Shanghai 200438, China

**Keywords:** exercise, diabetic cardiomyopathy (DCM), FGF21, irisin, GDF15, apelin, adiponectin, α-KG, M2 microphage, NK cells, Tregs, microRNAs, lncRNAs, PGC-1α, P2X7R, SIRT1, O-GlcNAcylation, intestinal flora

## Abstract

Diabetic cardiomyopathy (DCM) is one of the cardiovascular complications of diabetes, characterized by the development of ventricular systolic and diastolic dysfunction due to factors such as inflammation, oxidative stress, fibrosis, and disordered glucose metabolism. As a sustainable therapeutic approach, exercise has been reported in numerous studies to regulate blood glucose and improve abnormal energy metabolism through various mechanisms, thereby ameliorating left ventricular diastolic dysfunction and mitigating DCM. This review summarizes the positive impacts of exercise on DCM and explores its underlying molecular mechanisms, providing new insights and paving the way for the development of tailored exercise programs for the prophylaxis and therapy of DCM.

## 1. Introduction

Diabetes is a metabolic disorder marked by hyperglycemia, which is the third largest threat to human health. The global population of people suffering from diabetes is increasing rapidly and is expected to reach 783.2 million [1] by 2045. Cardiac complications are the most common factors of death and disability linked to diabetes. Diabetic cardiomyopathy (DCM) is defined as structural or functional cardiac impairment in the absence of overt heart disorders, such as coronary heart disease, hypertension, and valvular disease [2,3]. The initial phase of DCM is mainly the left ventricular diastolic dysfunction (LVDD), followed by decreased contractility, which will eventually lead to heart failure (HF) [4], resulting in a high chance of mortality. DCM is among the most severe complications associated with diabetes. The pathological features of DCM are energy metabolic disorders caused by high glucose, mitochondrial dysfunction, vascular lesion, cardiomyocyte fibrosis, oxidative stress, calcium imbalance, systemic and tissue inflammation, and so on [5,6,7,8,9,10].

At present, about 22% of diabetic patients over 64 years old face complications with cardiomyopathy. However, the main treatment is still limited to medication controlling blood sugar and blood lipids [11]. Exercise has been proven to enhance the health and life quality of patients with a wide range of diseases [12,13,14,15,16]. Regular exercise has been regarded as an effective non-drug means for the prevention and treatment of diabetes and its complications [14]. Numerous studies have reported that exercise is able to ameliorate DCM through diverse mechanisms, for instance, by adjusting the expression and activity of exerkines, M2 macrophages, intestinal flora, metabolites, and immune factors. This review aims to summarize the function and molecular mechanism of exercise in ameliorating DCM. It may offer novel insights into exercise-based approaches for the prophylaxis and therapy of cardiometabolic disorders.

## 2. Exercise-Mediated Amelioration of DCM in Human Studies

Numerous clinical studies verified that exercise is able to delay the progress of DCM. The LVDD is a remarkable characteristic and evaluation indicator of patients with DCM, which could be ameliorated by exercise intervention. In an observatory study including 18 male T2DM patients, echocardiography showed that the LVDD was improved after 12-week aquatic training, as indicated by marked suppression of E/E’ (the ratio between mitral E wave and E’ wave), which implied the improvement of LVDD [17].

A systematic review revealed that combined aerobic exercise training (AET) and resistance training (RT) exhibited more advantages for T2DM subjects in glycemic control than AET alone and RT alone, thereby more efficiently decreasing the risk of DCM [18]. Another meta-analysis of 2033 participants, among whom 7% had T2DM and 11% were overweight or obese, demonstrated that the high-intensity interval training (HIIT) group had better global insulin sensitivity than the continuous exercise group and the non-exercise group, thus reducing global insulin resistance (IR), which would improve the energy metabolism of cardiomyocytes to potentially function as an inhibitor of DCM [19].

MicroRNA (miRNA) is involved in the pathogenesis of DCM, and exercise can regulate the expression of miRNAs through various pathways to reduce the different pathological changes related to DCM. A study pointed out that, in patients with T2DM, the circulating level of miR-146a was downregulated, which led to the elevation of the expression of downstream pro-inflammatory cytokines, thereby aggravating inflammatory infiltration in the cardiomyocytes of patients with DCM [20,21]. However, aerobic exercise training increased the miR-146a serum level and alleviated inflammatory infiltration in cardiomyocytes [22]. In addition, miR-192 contributes to inhibiting the activity of the CXCL2 gene to exert an anti-inflammatory effect [23]. Circulating miR-192 levels were decreased in T2DM patients [24]. The upregulation of the miR-192 serum level was observed in healthy young adults that were subjected to an acute exercise trial [25]. Therefore, it is speculated that exercise is likely to regulate the serum levels of multiple miRNAs to reduce DCM-related cardiac inflammation and injury in humans.

Moreover, based on a 4-month moderate-intensity dance training (60 min/day, twice a week) experiment, Borges et al. found that, after dance training, the plasma levels of inflammatory cytokines were downregulated, and anti-inflammatory cytokines were increased in 10 patients with T2DM, compared to their levels in 12 healthy individuals [26]. Given that excessive inflammation response leads to the development of DCM, dance training may be an effective way for alleviating DCM.

In summary, mounting evidence supports that exercise can alleviate cardiac dysfunction in patients with DCM [22,25,26] (Figure 1A). The frequency, intensity, and type of exercise are regarded as important influencing factors [18,19]. Although HIIT has been proven to be more efficacious in reducing IR and enhancing energy metabolism, HIIT has some cardiovascular risks in patients with DCM compared to moderate-intensity exercise. Therefore, considering safety and stability, moderate-intensity exercise is more recommended. The optimal exercise protocol for patients with DCM remains to be deeply investigated.

## 3. Exercise-Mediated Alleviation of DCM in Animal Studies

The exercise-mediated alleviation of DCM has been demonstrated in numerous animal studies. For instance, an 8-week swimming training program (90 min/day, 5 days/week) was conducted in an experiment, which demonstrated that swimming was effective in increasing the myocardial contractile capacity of STZ-induced diabetic Wistar rats, as evidenced by the increase in cardiomyocyte volume and width [27]. The contractile capacity of cardiomyocytes, which depends on myocardial conformation and structure, closely relates to the cardiac function. On the other hand, myocardial contractile function is also influenced by the Ca^2+^ homeostasis of cardiomyocytes [28]. In a T2DM mouse model, Na^+^-Ca^2+^ exchange was suppressed, reducing the concentration of Ca^2+^ in cardiomyocytes and the accumulation of Ca^2+^ in cytoplasm, along with a reduction in SERCA2a gene expression, blocking the entry of Ca^2+^ into the sarcoplasmic reticulum (SR). These changes directly led to myocardial systolic dysfunction. On the contrary, exercise intervention has been verified to reverse Ca^2+^ imbalance. A study by Stolen [29] proved that myocardial contractility could be promoted in db/db mice after 13-week HIIT (80 min/day, 5 days/week), which involved the restoration of L-type Ca^2+^ channels, improved Na^+^-Ca^2+^ exchange as well as the upregulation of SERCA2a expression, and activity in cardiomyocytes.

Abnormal myocardial glucose metabolism leads to myocardial energy metabolism disorder, which causes damage to the cardiac structure and function [30,31], eventually inducing DCM. Broderick found that increased rates of myocardial glucose oxidation and utilization linked to enhanced left ventricular function were detected in STZ-induced SD rats after physical training on a rodent treadmill set at 8° for 10 weeks (the speed gradually raised from 22 m/min to 32 m/min and the duration increased from 10 min to 60 min), which indicated the effect of exercise on protecting the heart of mice with DCM [32].

Myocardial fibrosis, caused by extracellular matrix (ECM) deposition, serves as a crucial pathological characteristic of DCM, which will bring about cardiac remodeling through collagen accumulation, interstitial fibrosis, and perivascular fibrosis [33,34,35]. Inversely, HIIT was conducive to reducing cardiac fibrosis and inhibiting the exacerbation of DCM in the model of T1DM SD rats induced by alloxan. Before HIIT, the treadmill exercise experiment with increasing speed was carried out to determine the exercise protocol. The speed was gradually raised from 10 m/min every 3 min until reaching the maximum exercise speed, which was then kept for at least 3 min. In the above exercise protocol, 80% of the maximum speed and 80% of the time taken to reach that speed are applied as the final experimental scheme [36].

Previous studies have shown that, in diabetes, cardiac inflammation will be aggravated, leading to a series of pathological changes, including myocardial fibrosis and myocardial hypertrophy, which participates in the progression of DCM. It was verified that the upregulation of NLRP3 protein expression in the cardiomyocytes of mice with DCM induced by HFD was observed to promote the formation and activation of inflammasome, which is represented by the increased expression of caspase-1 and IL-β, and these changes could be inhibited by 20-week MIT (5 days/week), accompanied by the improvement of cardiac diastolic dysfunction [37]. Oxidative stress is regarded as a crucial factor in the occurrence of DCM. In the physiological state, moderate oxidative stress may increase the activity and function of proteins, but excessive oxidative stress will cause the interaction of reactive oxygen species with lipids, proteins, and DNA, resulting in pathological changes [38,39]. Hyperglycemia directly facilitates the generation of abundant reactive oxygen species (ROS) and brings about oxidative stress in cardiomyocytes, thus inducing cardiomyocyte apoptosis and cardiac dysfunction. Gimenes et al. [40] constructed an STZ-induced diabetic Wistar rat model and observed that, compared to the non-exercised diabetic group, the group with 9-week low-intensity physical exercise (11 m/min, 18 min/day, 5 days/week of treadmill running) had lower ROS production and higher activity of antioxidant enzymes in the cardiomyocytes of rats, which reduced the oxidative stress response of cardiomyocytes. In addition, echocardiography showed that the left atrial diameter of the exercised group was smaller than that of the non-exercised group, suggesting that low-intensity exercise improved diastolic dysfunction. Besides, it is confirmed that endothelial nitric oxide synthase (eNOS) and manganese superoxide dismutase (MnSOD) are significant effectors for decreasing the oxidative stress of endothelial cells in the diabetic heart [41]. A 7-week MIT (60 min/day, 5.2 m/min) experiment was conducted on db/db mice. On the one hand, moderate-intensity training (MIT) promoted the production of MnSOD in mitochondria of aortic endothelial cells, which inhibited the activity of free radicals and alleviated oxidative stress. On the other hand, MIT increased the eNOS protein expression in endothelial cells, facilitated the generation of NO, and then, improved endothelial vasodilation function [42]. With the progression of diabetes, the aortic endothelial cells in the heart will gradually lose normal diastolic function due to excessive oxidative stress response, eventually leading to cardiac dysfunction. Therefore, exercise plays an important role in alleviating DCM by reducing oxidative stress in cardiac endothelial cells.

Mitochondria are essential for energy metabolism in cardiomyocytes. Many studies have demonstrated that the mitochondrial dysfunction of cardiomyocytes, such as decreased mitochondrial density, increased mitochondrial matrix, and defective mitochondrial biogenesis, is the important pathogenesis of DCM [33,43,44,45]. It was verified that, in the cardiomyocytes of db/db mice, the mRNA expression of PGC-1α was decreased, which inhibited the mRNA expression of its downstream transcription factors mitochondrial transcription factor A (TFAM), mitochondrial transcription factor B2 (TFB2M), and nuclear respiratory factor 1 (NRF1), resulting in the impaired mitochondrial biosynthesis of cardiomyocytes, thereby inducing DCM. However, 15-week treadmill training at a speed of 10 m/min for 1 h/day significantly ameliorated the above mitochondrial dysfunction and exerted the cardiac protective effect [32]. In another study, a 5-week MIT program on db/db mice at a speed of 10–11 m/min was carried out, and it is found that exercise reduced the level of myocardial dynamin-related protein 1 (Drp1), which is a regulator of mitochondrial fission, along with the upregulation of mitochondrial transmembrane potential, thereby alleviating the excessive fission of cardiomyocytes mitochondria, improving mitochondrial function and preventing DCM-related cardiac remodeling [46].

Besides, in another study, STZ was used to induce T1DM in male SD rats, and those rats participated in a 4-week treadmill running program at a speed of 25 m/min and a duration of 60 min/day. The M-mode echocardiography observed the declined cardiac fractional shortening (FS%), which indicated that the impairment of DCM was partly reversed by treadmill running [17,47]. Research verified that voluntary running for 8 weeks upregulated the mRNA expression of cardioprotective genes, including *ANP*, *BNP*, *OP*, and *GATA4*, accompanied by reduced cardiac apoptosis in 5-week-old male ob/ob mice, which contributed to the improvement of DCM in these mice [48].

In conclusion, in DCM animal models, a variety of exercises are proven to help attenuate cardiac injury induced by hyperglycemia (Figure 1B). Specifically, exercise has been demonstrated to alleviate myocardial fibrosis [36], abnormal energy metabolism [32], inflammatory response [37], mitochondrial dysfunction [46], excessive oxidative stress [40,42], and other pathological processes [27,29] by acting on cardiomyocytes, endothelial cells, and other target cells, which ultimately normalizes cardiac systolic and diastolic functions and protects against DCM-related cardiac remodeling.

## 4. Molecular Mechanisms of Exercise-Induced Mitigation of DCM

### 4.1. The Contribution of Exercise-Induced Cytokines and Metabolites in the Protective Effects Against DCM

#### 4.1.1. Fibroblast Growth Factor 21 (FGF21)

FGF21, a cytokine that is predominantly secreted by the liver, has been shown to be involved in the modulation of metabolism through FGF receptor 1 (FGFR1) and its required coreceptor KLB (β-klotho) [49]. It was demonstrated that FGF21 has cardioprotective effects, including reducing angiogenesis, inflammatory responses, and lipid accumulation in the cardiomyocytes of diabetic mice [50,51]. In obese older adults and obese mouse models [52,53], chronic exercise significantly reduced circulating FGF21 levels, suggesting that FGF21 is a factor associated with exercise. Based on this, Leigang and colleagues explored the beneficial impact of FGF21 on the heart of DCM under exercise stimulation [54]. The researchers conducted a 4-week treadmill exercise experiment at a speed of 15 m/min, 1 h per day, 5 days a week in HFD-fed obese mice and observed that exercise markedly decreased the circulating FGF21 level and dramatically upregulated the KLB expression in cardiomyocytes of obese mice but did not affect the cardiac expression of FGFR1. This suggests that treadmill exercise increases cardiac sensitivity to FGF21 by elevating the myocardial expression of KLB. Then, DCM was induced by STZ in HFD-fed mice. In diabetic conditions, cardiomyocytes exhibited damaged mitochondrial morphology and function, manifested by mitochondrial swelling, irregularities, the absence of cristae, decreased ATP production, the downregulated activity of mitochondrial respiratory chain (MRC) complex II and V, the hyperacetylation of succinate dehydrogenase complex flavoprotein subunit A, complex II subunits (SDHA) and ATP synthase subunit A, complex V subunit (ATP50), and the significantly increased acetylation of total mitochondrial protein in cardiomyocytes. Further studies showed that, in the hearts of diabetic mice, the acetylation levels of long-chain acyl-CoA dehydrogenase (LCDA), a key enzyme catalyzing the first step of mitochondrial FAO, and superoxide dismutase 2 (SOD2), a crucial mitochondrial antioxidant enzyme, were significantly increased by about three times in cardiomyocytes, while treadmill exercise only reversed the hyperacetylation of LCDA and SOD2 in the hearts of diabetic wild type (WT) mice but not in FGF21 knockout (KO) mice. These findings imply that FGF21 functions as a downstream effector of exercise to exert a protective impact on DCM by decreasing the hyperacetylation of myocardial mitochondrial enzymes to increase their enzyme activities. Sirtuins are deacetylases that can regulate mitochondrial enzyme activity and mitochondrial function [55]. In this study, only the cardiac mRNA expression of SIRT3, but not those of other sirtuins, was obviously inhibited in diabetic mice compared to healthy sedentary mice. The poor responsiveness of FGF21 KO mice to exercise on the improvement of myocardial function was reversed after the overexpression of cardiac SIRT3 in cardiomyocytes, contributing to the mitigation of mitochondrial damage caused by diabetes and cardiac dysfunction, as indicated by increased LCAD and SOD2 activities and the deacetylation of SDHA and ATP50. In the in vitro experiment, doxycycline was administered to human-induced pluripotent stem cell-derived cardiomyocytes (hiPSC-CMs) to trigger the expression of KLB, which mimics the impact of exercise on increasing cardiac KLB expression. In hiPSC-CMs with overexpressed KLB, the phosphorylation of FOXO3 at Ser413 was detected when treated with recombinant human FGF21 (rhFGF21) protein, associated with the upregulated mRNA expression of SIRT3. Chromatin immunoprecipitation assays further exhibited that FOXO3 directly bound to cis-elements between −904 bp and −896 bp of SIRT3 gene promoter, promoting the mRNA transcription of SIRT3. FGF21 is known to activate AMPK signaling after binding to its receptor, and AMPK can directly phosphorylate FOXO3 at Ser413. In hiPSC-CMs treated with FGF21, transfection with dominant-negative AMPK-α-2 (D159A) (dnAMPK) plasmid significantly inhibited FOXO3 phosphorylation at Ser413 and SIRT3 mRNA expression. An in vivo experiment also reached a consistent conclusion. Exercise obviously induced AMPK phosphorylation at Thr172 in the cardiomyocytes of WT mice but not in FGF21 KO mice. Therefore, exercise promotes the binding of FGF21 to KLB on cardiomyocytes, subsequently activates AMPK-FOXO3 signaling, upregulates the mRNA expression of cardiac SIRT3, and reduces the hyperacetylation of myocardial mitochondrial enzymes to improve mitochondrial function, thus alleviating DCM-related heart damage.

Cardiomyocytes are not only targets of FGF21 but also one of the cell types that can secrete FGF21 [56]. The FGF21 receptor FGFR1 is highly expressed in cardiomyocytes. Based on this, another study in a mice model with myocardial infarction (MI) investigated that 6-week aerobic exercise (10 m/min, 55 min/day, 5 days/week) elevated the protein expression of FGF21 and FGFR1 in cardiomyocytes, as well as decreased oxidative stress and apoptosis. This may be achieved by activating the PI3K-Akt signal to inhibit the expression of lysocardiolipin acyltransferase 1 (ALCAT1) [57]. ALCAT1 is an enzyme that triggers the pathological remodeling of cardiolipin. Cardiolipin, in turn, is indispensable to preserving the structural and functional stability of mitochondria [58]. In the hearts of MI, ALCAT1 exacerbates the abnormal modification of cardiolipin, thereby promoting oxidative stress and activating apoptotic pathways, leading to myocardial injury [57]. Therefore, it is indicated that FGF21 binds to FGFR1 under exercise stimulation and then activates PI3K-Akt signaling, which inhibits the expression of ALCAT1 to alleviate apoptosis and oxidative stress in the cardiomyocytes of mice with MI [57]. Because apoptosis and oxidative stress may facilitate the progression of DCM, exercise may improve the cardiac function of individuals with DCM through the FGF21-regulated PI3K-Akt- ALCAT1 axis, which remains to be further explored.

Collectively, FGF21, as a regulatory cytokine for metabolism, may serve as a significant downstream effector of exercise to alleviate DCM (Figure 2A), and further studies are required to dissect the underlying molecular mechanisms.

#### 4.1.2. Irisin

Irisin, a myokine found in recent years, is a polypeptide that consists of 112 amino acids and is clipped from fibronectin type III domain-containing protein 5 (FNDC5) by proteolytic enzyme [59]. Irisin works to improve metabolic function through binding to its receptor integrin αV/β5 and is upregulated under exercise stimulation. Additionally, irisin and FNDC5 are highly expressed in the heart and exert a cardioprotective effect by diminishing oxidative stress, alleviating cardiac apoptosis and fibrosis [60,61]. However, in the heart tissues of T2DM mice, irisin expression decreased. According to recent studies, serum irisin, the expression of FNDC5/irisin genes and proteins, as well as the downstream AMPK activity in cardiomyocytes were significantly increased after 8 weeks of aerobic exercise in rats with DCM, which were closely related to the decreased mRNA and protein expression of mitochondrial fission proteins, such as Drp1, fission-1 protein (Fis1) and mitochondrial fission factor (MFF) in cardiomyocytes, and the amelioration of myocardial fibrosis [62]. However, these protective effects by exercise were abolished after the injection of Cyclo RGDyk (an inhibitor of irisin receptor αV/β5), along with a decrease in the AMPK phosphorylation level [62]. It is known that, under the condition of hyperglycemia, the excessive mitochondrial fission of cardiomyocytes will damage the structure and function of mitochondria, which results in the death of cardiomyocytes due to energy metabolism disorders and, ultimately, leads to the deterioration of cardiac function [63]. In addition, AMPK has been demonstrated to be activated and phosphorylated by irisin in previous studies [64] and takes part in the mitigation of mitochondrial dysfunction caused by hyperglycemia [64]. Consequently, it is implied that the benefit of exercise on DCM-associated cardiac dysfunction may be achieved by upregulating the expression of irisin in cardiomyocytes, thereby activating the AMPK signal to reduce the expression of mitochondrial fission proteins. Another study reported that regular exercise significantly reduced IR, abnormal glycolipid metabolism, and inflammatory infiltration in the skeletal muscle of T2DM rats, which may be partly achieved by improving skeletal muscle metabolism through the exercise-induced activation of AMPK signal by irisin [65]. Since skeletal muscle is the largest metabolic organ, improvements in its metabolism in the diabetic state would contribute to the improvement of systemic metabolism, which may help to ameliorate DCM.

Zhou et al. [66] found that exercise-induced FNDC5/irisin was able to accelerate neovascularization and reduce cardiac fibrosis via promoting the repair of damaged cardiomyocytes by Nkx2.5^+^ cardiac progenitor cells (CPCs), which was associated with the elevation of proliferation marker Ki67, phosphorylated histone 3, as well as p38 acetylation, and the significant reduction in histone deacetylase 4 (HDAC4) in CPCs. In conclusion, it is suggested that upregulated irisin may contribute to protecting against DCM by acting on CPCs under exercise stimulation. Therefore, exercise promotes the upregulation of irisin, which can act on different types of cells in the heart and facilitates the amelioration of DCM (Figure 2B).

#### 4.1.3. Growth Differentiation Factor 15 (GDF15)

GDF15 belongs to the TGF-β family and is highly expressed in cardiomyocytes. GDF15 participates in inhibiting the body’s inflammatory response, extracellular matrix deposition, abnormal energy metabolism, and other pathological processes [67]. Normally, the serum levels of GDF15 are low, but they are increased significantly in diseases such as diabetes. For instance, serum GDF15 levels were elevated in asymptomatic patients with T2DM compared to the control group, and the patients with T2DM showed the deterioration of heart diastolic function [68], suggesting that GDF15 can be used as a reflection of cardiac function. GDF15 has an essential mitigation effect on DCM. After the global knockout of GDF15 in T2DM mice, the expression of myocardial glucose transporters (GLUT1, GLUT2) was reduced, triggering IR and decreased glucose tolerance in cardiomyocytes, thus exacerbating the DCM-related deterioration of cardiac function [69]. However, the expression of GDF15 can be increased by exercise stimulation. In a study involving 24 obese old adults, 12 months of aerobic exercise increased serum GDF15 levels, which was accompanied by a decrease in fat mass and improved whole body fat oxidation and insulin sensitivity [70]. This result indicates that exercise can induce an increase in the circulating level of GDF15 in elderly obese adults. However, whether exercise can increase the serum level of GDF15 in diabetic patients remains to be further investigated. In another exercise experiment, the serum GDF15 concentration of mice in the one-time acute running group increased immediately after exercise compared to sedentary mice. Therefore, it is believed that exercise increases the serum GDF15 level, which may improve global IR, thereby alleviating DCM. In conclusion, GDF15 may contribute to the exercise-induced mitigation of DCM (Figure 3A), which warrants further investigation.

#### 4.1.4. Apelin

Apelin serves as a peptide ligand for the G protein-coupled receptor known as APJ and acts through binding to APJ [71]. It is secreted by adipose tissues and is widely distributed in cardiomyocytes, vascular endothelial cells, skeletal muscle cells, and adipocytes [72,73]. Apelin has beneficial effects, such as improving systolic and diastolic function of blood vessels and promoting angiogenesis and energy metabolism. Compared to healthy individuals, patients with T2DM have lower serum apelin levels, accompanied by cardiac remodeling, which is primarily concentric left ventricular remodeling [74]. Therefore, it can be assumed that apelin may be conducive to protecting against cardiac dysfunction in a diabetic state.

Impaired angiogenesis is among the primary complications associated with diabetes, and microvascular dysfunction in DCM is known to cause cardiac hypertrophy and accelerate cardiac remodeling. Zeng and colleagues found that the expression of apelin and APJ was downregulated in the cardiomyocytes of diabetic db/db mice [75]. The systemic overexpression of apelin in db/db mice promoted cardiac angiogenesis and increased blood vessel density, manifested by the elevation of vascular endothelial growth factor (VEGF) and its receptor VEGFR2 and Angiopoietin-1 (Ang-1) and its receptor Tyrosine kinase receptor with immunoglobulin-like and epidermal growth factor-like domain-2 (Tie-2) in the heart tissues. At the same time, ROS production and the apoptosis of cardiomyocytes were decreased, accompanied by a marked enhancement in myocardial function, which was represented by elevated cardiac ejection fraction (EF%). These changes are regarded to be associated with the elevation of myocardial SIRT3 expression. In mice with DCM and systemic SIRT3-KO, apelin-induced angiogenesis and the upregulation of the expression of VEGF/VEGFR2 proteins were inhibited. The apelin-mediated increase in the protein expression of Ang-1/Tie-2 and the phosphorylation of the downstream target Akt were also suppressed when SIRT3 was knocked out [75]. Therefore, the cardioprotective function of apelin on DCM heart may be accomplished by facilitating SIRT3-dependent cardiac angiogenesis. In addition, exercise is able to reverse the low cardiac apelin and APJ expression caused by hyperglycemia. Because SIRT3 has been confirmed to be involved in exercise-mediated DCM alleviation, it is speculated that apelin/SIRT3 axis-mediated cardiac angiogenesis may contribute to the alleviation of DCM by exercise, which merits further investigation.

Sabouri et al. pointed out that, after 8 weeks of HIIT and MIT, the protein expression of apelin and APJ in the left ventricle of T2DM rats was significantly upregulated [76]. Meanwhile, the researchers detected increased NO protein expression and improved myocardial vasodilation function, which suggests that apelin may participate in exercise-regulated DCM remission by triggering the release of NO in the cardiac endothelial cells of heart tissues to reduce cardiovascular damage induced by hyperglycemia. Another exercise experiment revealed that, compared to sedentary rats with T1DM, the aerobic exercise group of diabetic rats had higher serum apelin levels and apelin mRNA expression in adipose tissues. This change was inversely correlated with systemic IR levels, indicating that apelin may be beneficial for decreasing myocardial energy metabolism disorders in DCM via enhancing insulin sensitivity and promoting glucose metabolism [77].

In summary, exercise promotes the expression and secretion of apelin, and then, apelin binds to APJ receptors on cardiomyocytes, cardiovascular endothelial cells, and other myocardial cells to improve glucose metabolism and cardiovascular function in diabetic conditions, thus alleviating DCM (Figure 3B).

#### 4.1.5. Adiponectin (APN)

APN, a kind of adipokine derived from adipocytes, has the function of improving systemic insulin sensitivity, promoting glucose and lipid metabolism, and lowering blood glucose, and its dysregulation is implicated in the pathogenesis of DCM [78]. Kim et al. fed db/db mice with a diet containing APN receptor agonist for 4 weeks to explore the protective impact of APN and its receptor on DCM [79]. The findings demonstrated that, compared to the control group, the serum APN level of mice fed with APN receptor agonists was increased, the systemic IR level was decreased, and the cardiac function was improved, including the reduction in cardiac fibrosis, inflammation, and apoptosis. This indicates that APN is beneficial for protecting the diabetic heart. APN has two receptors, adipoR1 and adipoR2, which can activate the AMPK signaling of cardiomyocytes and vascular endothelial cells upon binding with APN [80]. On the one hand, AMPK signaling subsequently upregulated the expression of PGC-1α in the cardiomyocytes and enhanced the function of myocardial mitochondria to improve metabolism. On the other hand, in cardiac vascular endothelial cells, the AMPK-Akt signal was activated to phosphorylate FOXO1 and then weakened its activity, leading to a marked elevation in the gene expression of downstream eNOS and, thus, alleviating vascular oxidative stress, which helps to improve the diastolic function of cardiac vessels in DCM. In addition, a randomized controlled trial of 22 patients with T2DM showed that serum APN levels were significantly higher in those who performed moderate-intensity treadmill running for 4 weeks compared to non-exercise controls [79]. Therefore, APN is considered as an exercise-related downstream mediator to alleviate DCM (Figure 4A).

#### 4.1.6. Alpha-Ketoglutarate (α-KG)

α-KG is a key metabolite in the TCA cycle, which exists in cytoplasm, mitochondria, as well as blood and acts as a crucial factor in modulating ATP production, oxidative stress, energy metabolism, and epigenetic modification [81,82,83,84]. Previous reports verified that the level of α-KG was downregulated in the cardiac mesenchymal cells of T2DM patients and heart tissues of HFD-fed mice [85,86], suggesting a potential involvement of α-KG level in the pathological changes in DCM.

Hyperglycemia alters the structure and transcriptome of myocardial chromatin, bringing about the abnormal activation of many signaling pathways in the heart and inducing cardiac dysfunction [87]. The epigenetic modification of the genome exerts a vital role in transcriptional reprogramming [88]. A latest investigation revealed that the changes in methylation and hydroxymethylation modifications occurred on genomic DNA in the left ventricular tissue of rats with STZ-induced DCM compared to controls. α-KG facilitates DNA demethylation and, thus, alleviates cardiac dysfunction caused by hyperglycemia. Mechanistically, α-KG promoted the binding of ten-eleven translocase enzyme (TET1) to the upstream elements of TGFBR2 and TGFBR3 genes, respectively, which reduced methylation and hydroxymethylation enrichment in the intron region of the pro-fibrosis genes TGFBR2 and TGFBR3 to downregulate their transcriptional expression, thereby reversing cardiac fibrosis in rats with DCM. In addition, α-KG also increased the binding of the TET to thymine DNA glycosylase (TDG) to form the TET1-TDG complex, thus promoting the oxidation of 5′ methylcytosine (5 mc) to cytosine in the diabetic heart, and thereby removing methylation and hydroxymethylation modifications in the left ventricular genome of DCM and restoring the cardiac function [89]. An experiment on obese mice fed with HFD exhibited that, compared to the non-exercise group, treadmill exercise induced a time-dependent rapid increase in serum α-KG levels [90], which implies that exercise can upregulate the serum level of α-KG. However, it remains unknown whether such exercise-induced changes in α-KG level occur under diabetic conditions. Another study found that both acute resistance exercise and short-term HIIT induced a marked elevation in serum α-KG levels in healthy people [91]. In conclusion, α-KG may function as a metabolite in reaction to exercise and help with the alleviation of DCM (Figure 4B), which merits further investigation.

### 4.2. The Involvement of Immune-Related Factors in the Amelioration of DCM by Exercise

#### 4.2.1. M1/M2 Macrophages

DCM is a pathological process with chronic inflammation. Macrophage-mediated inflammation persists throughout the course of DCM. Compared to control mice, the ratio of M1-type macrophages in the heart tissues of T2DM mice was significantly increased, while the ratio of M2-type macrophages was markedly inhibited, which promoted the occurrence of DCM by inducing myocardial inflammation [92]. The polarization of macrophages from M1-type to M2-type diminishes inflammation in cardiomyocytes, thus alleviating myocardial injury. Miao et al. treated rat bone marrow mononuclear cells (BMMCs) with granulocyte/macrophage colony stimulating factor (GM-CSF/M-CSF) to induce the differentiation into macrophages and co-cultured them with H9C2 cells to study the effect of macrophages on cardiomyocytes [93]. Immunofluorescence results showed that macrophages treated with high glucose exhibited an increased expression of α-SMA and fibronectin-1 (FN-1) in H9C2 cells, contributing to myocardial fibrosis. Therefore, high glucose significantly increases the polarization of M1 macrophages, triggering myocardial cell inflammation to promote fibrosis. However, exercise has been proven to elevate the proportion of M2 macrophages. Eight-week eccentric exercise significantly promoted M2 macrophage polarization in obese mice [94], which indicates that exercise helps to promote the polarization of M2 macrophages in the obese state. However, whether this effect of exercise exists in the diabetic model remains to be further explored (Figure 5A).

#### 4.2.2. Regulatory T Cells (Tregs)

Tregs are T cell subsets that inhibit inflammatory response and maintain immune homeostasis. In addition to regulating the initiation and role of immune cells, Tregs are able to gather in parenchymal tissues and then promote the repair of damaged sites [95,96]. The proportion of circulating Tregs in diabetic patients decreased significantly [97]. It was reported that db/db mice in the high-dose Tregs group demonstrated significant amelioration in cardiac systolic and diastolic function compared to the control group, along with reduced cardiac hypertrophic gene expression and fibrosis [98], indicating that Tregs play an important role in alleviating DCM. Twelve weeks of Tai Chi exercise was shown to improve the glucose metabolism of patients with T2DM to reduce the glycosylation modification of Tregs, ultimately increasing the immune function of Tregs, thereby reducing the risk of concurrent DCM [99]. This may be because, after exercise, the glycosylation modification of Tregs is reduced, the activity of anti-inflammatory factors secreted by Tregs is elevated, or the pathways related to survival in Tregs cells, such as the PI3K-Akt signal, are enhanced. Therefore, it is speculated that exercise may ameliorate DCM by Tregs to reduce cardiac injuries (Figure 5B). The effect and molecular mechanism of exercise-mediated Tregs in alleviating DCM requires more in-depth investigation.

#### 4.2.3. Natural Killer Cells (NK Cells)

NK cells are a kind of lymphocyte that can recognize and attack tumor cells and virus-infected cells. The cytotoxic activity is an important indicator of their functional status [100]. DCM is associated with a decrease in NK cell cytotoxic activity. Kim et al. established T1DM and T2DM mouse models and isolated NK cells from the spleen [101]. NKp46 is known to be a representative activating receptor of NK cells with cytotoxic activity [102]. Flow cytometry results exhibited that compared to normal mice, the NKp46 expression on the surface of the NK cells of diabetic mice was significantly decreased, and the cytotoxic activity of the NK cells of T1DM mice gradually reduced with the progression of diabetes, which indicated that diabetes would progressively impair the cytotoxic activity of NK cells, which was not conducive to the clearance of injured cells and, thus, increased the risk of complications with DCM. NK cells can attenuate inflammation and IR in mouse models of T2DM [103]. A 4-week treadmill exercise experiment on rats with T1DM verified that the proportion of serum NK cells in diabetic rats were partially increased, the cytotoxic activity of NK cells damaged by hyperglycemia was elevated to a certain extent, and immune function was also upregulated after exercise, which suggests that the elevation of the cytotoxic activity of NK cells is involved in exercise-mediated DCM improvement [104]. In conclusion, the molecular mechanism by which exercise alleviates DCM may be intimately linked to the increase in the cytotoxic activity and immune function of NK cells (Figure 5C), which needs further exploration.

### 4.3. The Protective Effect and Mechanism of Exercise-Regulated miRNAs on DCM

MiRNA constitutes a category of non-coding single-stranded RNA molecules encoded by endogenous genes and is engaged in the pathological process of DCM, affecting cardiac function by regulating cardiomyocyte metabolism, apoptosis, and inflammatory response [105]. MiR-206 promotes apoptosis in cardiomyocytes and negatively regulates its downstream target heat shock protein 60 (HSP60). HSP60, located in cytoplasm and mitochondria, is a mitochondrial chaperone protein with anti-apoptotic effects. HSP60 in the cytoplasm can bind to pro-apoptotic proteins, including Bax and Bak, and then inhibit their entry into the mitochondrial membrane, thus inhibiting apoptosis [106]. It was found that the expression of miR-206 was upregulated in the left ventricular tissues of T1DM rats, accompanied by increased myocardial apoptosis. However, HIIT and MICT has been proven to reverse this change. Exercise inhibited the expression of miR-206 in left ventricular tissues and then elevated HSP60 protein expression to suppress the apoptosis of ventricular myocytes [107]. Therefore, it is likely that downregulated miR-206 is involved in the course of the exercise-induced alleviation of DCM by increasing the protein expression of HSP60.

MiR-486-5p functions as a cardioprotective factor that has been shown to inhibit apoptosis [108,109]. The expression of miR-486-5p was suppressed in the heart tissue of T1DM mice, leading to the apoptosis of cardiomyocytes. However, 12-week treadmill exercise significantly upregulated the expression of miR-486-5p in heart tissues, with a reduction in left ventricular end-diastolic diameter (LVEDD) and left ventricular end-systolic diameter (LVESD), an increase in the EF%, and the remission of DCM cardiac dysfunction [110]. This was related to the inhibition of mammalian sterile 20-like kinase 1 (Mst1) expression. Mst1 is able to inhibit glucose metabolism, damage mitochondria autophagy, as well as induce apoptosis, and has been shown to participate in the process of MI and atherosclerosis [111,112,113]. Dual luciferase reporter gene assay proved that Mst1 was the downstream target gene of miR-486-5p. Exercise elevated miR-486-5p and then promoted its direct binding with Mst1 mRNA, thereby inhibiting the translation of Mst1 mRNA and reducing Mst1 protein expression. This mechanism helps alleviate apoptosis in DCM cardiomyocytes and promotes the improvement of cardiac function [110]. In summary, various miRNAs are regulated by exercise to participate in the exercise-mediated amelioration of DCM, which warrants further investigation.

MiR-1 is exclusively expressed in myocardium to induce apoptosis. The expression of miR-1 was elevated in the heart tissues of T1DM rats with DCM. Meanwhile, echocardiography showed that the LVEDD and the LVESD were seen to be enlarge, and the EF% was decreased, indicating the occurrence of cardiac dysfunction in diabetic conditions. However, compared to the non-exercise group, 5-week HIIT and moderate-intensity continuous training (MICT) significantly inhibited the expression of miR-1 in cardiac tissues and upregulated the mRNA expression of its downstream anti-apoptotic target insulin-like growth factor 1 (IGF-1) and its receptor IGF-1R, thereby inhibiting the apoptosis of cardiomyocytes and alleviating DCM, which indicates that the favorable impacts of exercise on DCM may be partly regulated by the downregulation of miR-1 expression [114].

MiRNAs have also been verified to be released into the circulation in the form of exosomes, which are also involved in the regulation of myocardial function. MiR-29b and MiR-455 are negative regulators of pro-fibrotic genes, such as matrix metalloprotease 9 (MMP9) and collagen I (Col1a1). The overexpression of miR-29b and miR-455 inhibits the synthesis of extracellular matrix and then reduces fibrosis. MiR-29b and miR-455 were observed to be significantly decreased in the myocardium of mice with T2DM [115]. Pankaj et al. [116] demonstrated that exosomes in the cardiac vascular wall of mice with T2DM were released into the serum, and exercise training promoted the release of more exosomes. Compared to the non-exercise group, the expression of miR-29b and miR-455 in the exosomes of the exercise group was significantly upregulated, subsequently binding to the 3′ region of MMP9 mRNA in heart tissues to decline the MMP9 protein expression, thereby inhibiting cardiac fibrosis.

Sarlak et al. [117] found that 10 weeks of swimming training combined with sodium butyrate treatment reduced the expression of miR-34a in cardiomyocytes and promoted cardiac angiogenesis in T2DM mice. Sodium butyrate is conducive to anti-inflammatory, blood glucose control, and glucose metabolism improvement, and it participates in the alleviation of cardiac dysfunction in T2DM mice [118,119]. MiR-34a is a downstream target gene of P53 and is involved in inducing apoptosis [120]. The high expression of miR-34a in cardiomyocytes inhibits the expression of sirtuin 1 (SIRT1) in endothelial cells and, thus, prevents angiogenesis. Mechanistically, compared to sodium butyrate therapy alone and swimming training alone, combination therapy significantly downregulated the expression of miR-34a in cardiomyocytes and upregulated the protein expression of SIRT1 in endothelial cells. Forkhead box O1 (FOXO1) and hypoxia-inducible factor 1 (HIF-1α) are two downstream targets of SIRT1. On the one hand, elevated SIRT1 induced FOXO1 deacetylation in endothelial cells and improved the transcriptional activity of FOXO1 to increase the expression of genes related to angiogenesis, thereby promoting the vascular tube formation. On the other hand, SIRT1 elevated the protein expression and nuclear translocation of HIF-1α in endothelial cells to promote the expression of its downstream target genes, such as VEGF, thereby increasing angiogenesis and alleviating the cardiac dysfunction of DCM. This study suggests that exercise combined with sodium butyrate therapy is important for inhibiting miR-34a in cardiomyocytes and promoting DCM cardiac angiogenesis (Figure 6A). The potential role of miR-34a/SIRT1 axis-mediated cardiac angiogenesis in the alleviation of DCM by exercise remains to be further studied.

### 4.4. The Function and Mechanism of Exercise-Regulated Long Non-Coding RNAs (lncRNAs) in DCM

LncRNA is a class of long-stranded non-coding RNAs regulating the expression of target genes at the transcriptional and post-transcriptional levels [121]. LncRNA is engaged in several physiological processes, among which are cell proliferation, differentiation, and metabolism. Metastasis-associated lung adenocarcinoma transcript 1 (MALAT1), also known as nuclear-enriched abundant transcript 2 (NEAT2), is an lncRNA that is located in the nucleosome and widely expressed throughout the body and has been shown to act as a sponge for miRNA to regulate miRNA translation [122]. In addition to improving insulin sensitivity and stimulating insulin secretion, MALAT1 KO also reduces hepatic steatosis and IR [123]. IR is an important pathogenesis of DCM, which damages the energy metabolism of cardiomyocytes and leads to cardiac dysfunction. In the T2DM mouse model [124], the serum level of MALAT1 was increased, and the IR index was higher than that of the healthy control group. This is associated with upregulated levels of resistin in the serum. Resistin is an adipocyte-secreted cytokine, which induces IR and glucose intolerance, impairing glucose homeostasis [125]. Swimming has been proven to reverse this phenomenon [124]. Mechanistically, exercise suppressed the level of MALAT1 in the circulation of T2DM mice and subsequently increased miR-382-3p to decrease the expression of resistin in serum. In vitro experiments showed that the overexpression of MALAT1 elevated the resistin level in HUVECs cultured with high glucose, while the co-transfection of miR-382-3p and pcDNA-MALAT1 abolished the elevation of resistin induced by MALAT1. Therefore, it is suggested that the downregulation of MALAT1 can participate in the exercise-induced amelioration of DCM by improving the energy metabolism of cardiomyocytes.

Previous studies have established that vascular endothelium is a significant target organ of IR. IR can damage vascular endothelium through the inflammation pathway, which leads to cardiac dysfunction in diabetic mice [126]. Liu et al. [127] performed lncRNA chip analysis on mice with IR induced by HFD and found that, compared to the non-exercise group of mice, the expression of LncRNA FR030200 and FR402720 in the aortic endothelium of mice with 12-week swimming exercise was significantly decreased, accompanied by the improvement of vascular endothelial structural integrity. According to KEGG and GO analysis, lncRNA FR030200 and FR402720 are most likely to affect the cell cycle, cell proliferation, and differentiation pathways in the context of cardiac endothelial damage associated with IR. Further studies verified that the decline in lncRNA FR030200 and FR402720 activated E2 promoter binding factor 1 (E2F1) under exercise stimulation, to downregulate the mRNA expression of its downstream target gene neuronatin (Nnat) and alleviate IR-induced vascular endothelial inflammation, thereby reducing the risk of IR-related cardiac injury. E2F1 is proven to have potential anti-inflammatory impacts and is helpful for reducing vascular endothelial injury. For instance, E2F1 can inhibit the expression of VCAM-1 and e-selection induced by TNF-α [128]. In addition, Nnat is an obesity-related gene that is widely present in metabolism-related tissues, which may be involved in the inflammatory response mediated by glucose metabolism [129]. Considering that the pathological process of diabetes is characterized by IR and vascular endothelial injury, it is hypothesized that exercise may decrease the risk of developing DCM by regulating some lncRNAs to improve IR, which deserves further investigation.

The lncRNA cardiac physiological hypertrophy-associated regulator (Cphar) is reported to regulate cell growth, hypertrophy, and apoptosis. The overexpression of Cphar protected hearts from MI injury and cardiac dysfunction. Gao et al. [130] found that, after 3 weeks of swimming training, Cphar expression was significantly increased in the myocardial cells of the mice with MI. DEAD-box helicase 17 (DDX17) was recruited by Cphar to bind the CCAAT/enhancer binding protein beta (C/EBP-β), which prevented the transactivation of activating transcription factor 7 (ATF7) by C/EBP-β, thereby inhibiting the expression of ATF7 in cardiomyocytes to suppress myocardial ischemia reperfusion injury and cardiac remodeling, which may contribute to alleviating DCM. The potential cardioprotective effect mediated by exercise-induced lncRNA Cphar is worthy of further study.

Therefore, lncRNA may play important roles in the exercise-mediated remission of DCM, which involves alleviating IR and inflammation, improving glucose metabolism and endothelial function to ultimately reduce cardiac dysfunction in DCM (Figure 6B). More investigations are necessary to further examine the potential molecular mechanisms.

### 4.5. The Role of Other Exercise-Related Effectors and Processes in Improving DCM

#### 4.5.1. PGC-1α

PGC-1α acts as a crucial regulator of cellular energy metabolism [131,132], which shows high expression in the skeletal muscle, heart, and liver [133]. Numerous studies verified that PGC-1α promoted mitochondrial function and reduced metabolic disorder in the diabetic heart [134]. Impaired mitochondrial biogenesis in diabetes is intimately linked to cardiac impairment, and PGC-1α helps to mitigate this injury. It is reported that the mRNA level of PGC-1α in the heart tissues of db/db mice was decreased, which was accompanied by mitochondrial mtDNA replication defect and ultrastructural damage to the mitochondria, leading to the apoptosis of cardiomyocytes. However, the 15-week treadmill exercise significantly upregulated the mRNA levels of PGC-1α and its downstream transcription factors NRD1, TFAM, and TFB2M in heart tissues, promoted the expression of genes associated with mitochondrial biogenesis, and improved myocardial mitochondrial function, thus alleviating the cardiac dysfunction of DCM [135]. Moreover, a study reported that, in mice with T2DM-induced DCM, exercise promoted the expression of PGC-1α in cardiomyocytes to improve mitochondrial dysfunction and energy metabolism, which enhanced cardiac function [136]. Generally, as a potential therapeutic target, PGC-1α is beneficial for reducing DCM cardiac injury by enhancing the mitochondrial biosynthesis of cardiomyocyte in the context of exercise intervention (Figure 7).

#### 4.5.2. Purinergic 2X7 Receptor (P2X7R)

P2X7R is not only an ATP-gated ion channel but also an inducer of inflammatory activation and widely exists in cardiomyocytes [137]. Previous study found that the P2X7R inhibitor alleviated ischemia-reperfusion damage and relieved cardiac fibrosis in mouse models via inhibiting NLRP3-IL-1 signaling [138,139,140]. It was found that the protein and mRNA expression of P2X7R were markedly increased in the heart tissues of STZ-induced T1DM mice and in H9C2 cells treated with high glucose, accompanied by cardiomyocyte apoptosis and cardiac dysfunction. Treatment with P2X7R inhibitors reversed this change, suggesting that the overactivation of P2X7R induces DCM [141]. To investigate more deeply the function of P2X7R in the exercise model of mice with DCM, another study constructed P2X7R KO mice, and these mice were treated with STZ and fed with HFD to establish the T2DM model. The findings indicated that, compared to the DCM+EX group, the DCM + EX + P2X7R KO group exhibited significantly lower levels of TGF-β protein in heart tissues and less cardiac fibrosis. This indicated that P2X7R deletion can facilitate the exercise-mediated alleviation of DCM by improving cardiac fibrosis. Moreover, in diabetic mice, exercise is able to suppress the expression of lncRNA MIAT and upregulate the expression of its downstream target miR-150. MIAT is known to have binding sites for miR-150 [142] and is involved in cardiac hypertrophy and fibrosis [143,144]. MiR-150 can alleviate cardiac fibrotic injury [145]. Compared to the DCM group, mice in the DCM + P2X7R KO group showed lower lncRNA MIAT expression and higher miR-150 expression [142]. Thus, it is speculated that the P2X7R/lncRNA MIAT/miR-150 pathway may be a downstream mechanism through which exercise ameliorates DCM. In summary, P2X7R is a potential therapeutic target for exercise to alleviate DCM via the improvement of myocardial inflammation, cardiac fibrosis, and cardiomyocyte apoptosis. However, the underlying molecular mechanism still deserves to be further examined (Figure 7).

#### 4.5.3. The Function of SIRT1 in Exercise-Regulated Amelioration of DCM

Sirtuins are engaged in the modulation of various biological processes, including cell proliferation, apoptosis, aging, and DNA repair [146]. SIRT1, a nicotinamide adenine dinucleotide-dependent deacetylase found in the nucleus and cytoplasm [147], has been recognized as a target for the alleviation of DCM. SIRT1 alleviates DCM by inhibiting transcription factors that impair cardiac function, such as NF-κB and FOXO3, and increasing the expression of some factors that protect cardiac function, such as eNOS, PGC-1α, and ERK1/2 [148,149]. Roshan Milani et al. [150] believed that SIRT1 promoted cardiac endothelial growth and improved the ability of cardiac angiogenesis in patients with DCM. They found that SIRT1 protein expression and cardiac angiogenesis were reduced in the heart tissues of rats with DCM induced by STZ. Compared to the control group, 4-week treadmill exercise significantly upregulated the expression of SIRT1 protein in the heart tissues of rats with DCM, and the results of PECAM1/CD31 immunofluorescence staining showed that the cardiac angiogenesis was increased, suggesting that exercise alleviated DCM at least partly through promoting angiogenesis via SIRT1. Another study using the MI model in T2DM rats showed that 8-week swimming exercise reversed MI injury compared to the non-exercise group, which was manifested as the improvement of myocardial inflammation and the increased serum levels of cardiac biomarkers. This may be because of the activation and phosphorylation of AMPK by swimming exercise, which increased the protein expression of its downstream target SIRT1 and subsequently elevated the protein expression of PGC-1α [151]. Therefore, these results suggest that SIRT1 can react to exercise stimulation to improve cardiac function via multiple pathways, thereby alleviating DCM (Figure 7).

#### 4.5.4. The Function and Mechanism of Exercise-Induced O-GlcNAcylation in DCM

O-GlcNAcylation represents a post-translational modification process, where O-GlcNAc transferase (OGT) facilitates the attachment of UDP-GlcNAc to the serine and threonine residues of proteins. O-GlcNAcylation is found to participate in many biological processes, such as energy metabolism, signal transduction, and gene expression [152]. It has been found that the level of O-GlcNAcylation of total proteins in the myocardium of diabetic patients was higher than that of non-diabetic patients and was closely connected with left ventricular dysfunction [153]. In a non-diabetic mouse model, it was evidenced that the overexpression of OGT led to cardiac remodeling, manifested as cardiac fibrosis, the increased expression of pro-hypertrophic genes, and LVDD, presenting DCM-like pathological features. This implies that the elevated level of OGT in the diabetic heart would impair cardiac function and induce DCM. Further studies have revealed that OGT-induced damage to the heart may be due to the increased O-GlcNAcylation of two kinases PI3K and Akt, which could inhibit their phosphorylation to restrict their activities, thus inducing cardiac dysfunction [153]. In a rat model with hypertension, 12-week treadmill exercise has been verified to elevate the cardiac expression and phosphorylation levels of PI3K and Akt proteins and then activate the expression of downstream anti-apoptotic proteins, thereby playing a protective role in the heart [154]. Apoptosis is among the pathological traits of DCM. Another study pointed out that healthy mice in the swimming group had significantly reduced cardiac OGT mRNA expression and total O-GlcNAcylation levels of the proteins, accompanied by improved cardiac systolic function, compared to mice in the non-exercise group [155]. Therefore, it is speculated that exercise-regulated OGT and O-GlcNAcylation may contribute to the alleviation of DCM by exercise. In conclusion, exercise-regulated activity and the expression of cardiac OGT may be of great importance in reducing DCM via exercise, which needs further study (Figure 7).

#### 4.5.5. The Beneficial Role of Intestinal Flora in Exercise-Regulated Alleviation of DCM

Exercise can regulate the composition and function of intestinal flora [156,157,158]. By enriching the original dominant flora and potential probiotics, exercise participates in the prophylaxis and therapy of diabetes mellitus and its complications [8,159]. The number of *Veillonella* increased after exercise, which decomposed lactic acid into short-chain fatty acids (SCFAs), acetate, and propionate in the intestine [160]. Propionate is proven to promote the secretion of peptide YY and glucagon-like peptide-1 (GLP-1), which are essential for maintaining energy and glucose homeostasis in diabetic patients [161], thereby delaying the progression of DCM. Additionally, the replication rate of *Prevotella* decreased after exercise to weaken the biosynthesis of branched chain amino acids (BCAAs), which inhibited BCAAs/mTORC1-mediated IR [156] and may help to prevent the occurrence of DCM. *Anaerostipes* [162], *Lachnospiraceae* [163], and *Faecalibacterium prausnitzii* [164] were significantly enriched after exercise, which directly or indirectly produced SCFA to ameliorate inflammation. The levels of 5-aminopentanoic acid (produced by *Faecalibacterium prausnitzii*) and L-Norvaline (produced by *Bacteroides* and *Lactobacillus*) [165] were increased in diabetic rats after exercise, and they played beneficial roles in promoting insulin sensitivity and controlling blood glucose. The contribution of intestinal flora and their metabolites to the exercise-mediated alleviation of DCM and the underlying mechanism are worthy of further investigation (Figure 7).

## 5. Discussion

### 5.1. The Pathophysiology of Cardiac Remodeling in DCM Leads to the Echocardiographic Alterations Detectable via Tissue Doppler Imaging (TDI)

During the progression of DCM, left ventricular remodeling often occurs. It refers to the structural and functional changes in the left ventricle, jointly driven by pathological myocardial cell hypertrophy, apoptosis, myofibroblast proliferation, and interstitial fibrosis [166]. Based on TDI-detected echocardiographic changes, mice with DCM induced by both T1DM and T2DM develop LVDD, shown as an increased ratio of the early diastolic mitral flow velocity (E wave) to the early diastolic mitral annulus tissue velocity (E’ wave) [167]. This may be due to the severe damage to cardiomyocytes caused by hyperglycemic and oxidative stress, leading to myocardial diastolic dysfunction. At the cellular level, hyperglycemia disrupts the calcium handling process in cardiomyocytes. During diastole, the slow calcium efflux delays myocardial relaxation, preventing normal diastolic function [168]. Meanwhile, diabetes-induced myocardial fibrosis increases myocardial stiffness and reduces cardiac compliance, impeding normal heart diastole [169,170]. Consequently, the early-phase myocardial diastolic movement slows down, the E/E’ ratio rises, and diastolic function is impaired. Notably, the LVDD is more severe in mice with T1DM-induced DCM than those with T2DM-induced DCM. Exercise has been reported to partially restore the activities of the sarcoplasmic reticulum Ca^2+^ ATPase and Na^+^/Ca^2+^-exchanger in db/db mice [29] and reduce cardiac fibrosis in rats with DCM [36], which suggests that exercise can alleviate left ventricular remodeling in DCM.

### 5.2. The Similarities Between the Diabetic Heart and the Aging Heart

The aging heart is defined as a series of aging-associated degenerative changes in cardiac structure and function, like left ventricular hypertrophy and reduced diastolic function [171]. This shares similarities with the diabetic heart. Structurally, both the aging heart and diabetic heart show comparable alterations. Aging increases the thickness of the left ventricular wall and the left ventricular mass index (LVMI) and alters extracellular matrix composition [171]. In the diabetic heart, myocardial hypertrophy, interstitial fibrosis, and left ventricular remodeling are key pathologies. Moreover, both of them have vascular issues. Aging reduces elastic fibers and increases collagen in vessel walls [172], while diabetes, due to metabolic disorders, triggers cardiac vascular dysfunction, affecting cardiac blood supply. Functionally, they also have similar impairments of cardiac function. Diastolic function declines in both of them. In the aging heart, diastole is damaged by the reduced calcium-handling ability of aged cardiomyocytes [173]. In the diabetic heart, calcium homeostasis is disrupted by hyperglycemia, diastolic calcium efflux is slowed, and normal relaxation is hindered [168]. In addition, both will eventually gradually progress towards HF [174,175]. Notably, cellular senescence can be seen in the diabetic heart [175], suggesting a connection between the two. Moreover, like the diabetic heart, the aging heart can also be improved through exercise. Many studies show that exercise alleviates cardiac dysfunction in the elderly by reducing fibrosis, apoptosis, and cardiac remodeling [176,177]. In summary, these similarities may enhance our understanding of how metabolism and aging affect cardiac function.

## 6. Conclusions

In general, DCM, a common cardiovascular complication of diabetes, is a key hazard contributor for HF. Exercise proves to be an efficacious strategy for ameliorating cardiac function and averting HF among patients with DCM. Multiple investigations reveal that the cardioprotective effects of exercise can be attained via reducing IR and myocardial inflammation as well as improving glucose tolerance. In addition, a variety of animal experiments have been conducted to provide sufficient evidence for the exercise-mediated alleviation of DCM. The advantageous effect of exercise on relieving DCM involves various molecular mechanisms, and the target cells of exercise include cardiomyocytes, cardiac endothelial cells, cardiac fibroblasts, macrophages, and NK cells. Exercise is reported to regulate intestinal flora to maintain myocardial glucose homeostasis and promote microRNAs to enter the heart in the form of extracellular vesicles to play a cardioprotective role. Moreover, the adipokines, myokines, and metabolites released after exercise have been demonstrated to alleviate DCM. Although the ameliorative effect of exercise on DCM has been verified, the molecular mechanism still needs to be further explored.

Interestingly, some studies have pointed out that the impact of exercise upon DCM is also influenced by different forms, frequencies, and intensities of exercise. Exercise combined with drug intervention is another idea to improve DCM. Taken together, in the future, the optimal exercise regimen merits to be investigated further so as to provide more effective exercise-based approaches for alleviating DCM.

## Figures and Tables

**Figure 1 ijms-26-01465-f001:**
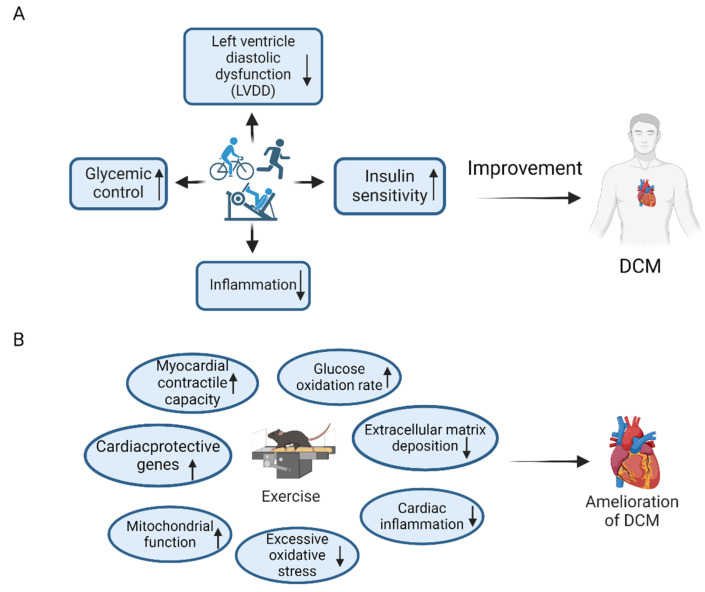
Exercise-mediated amelioration of diabetic cardiomyopathy (DCM). (**A**) Human studies have demonstrated that multiple types of exercise, including aerobic exercise and resistance exercise, can improve the cardiac function of patients with DCM and ameliorate the clinical symptoms of DCM through pathways such as reducing left ventricular diastolic dysfunction (LVDD), improving glycemic control, increasing global insulin sensitivity, and reducing chronic cardiac inflammation. (**B**) In animal studies, exercise has been proven to improve DCM through multiple pathways. For example, it can enhance cardiomyocyte contractility by strengthening Ca^2+^ homeostasis, increase glucose oxidation rates and utilization, reduce extracellular matrix deposition, and suppress cardiac inflammation to alleviate DCM-induced myocardial damage. Additionally, excessive oxidative stress in DCM heart is inhibited after exercise. Exercise improves mitochondrial function and then alleviates cardiac remodeling in DCM by increasing mitochondrial density and biogenesis, while reducing mitochondrial fission. Moreover, exercise enhances the expression of cardioprotective genes, which is beneficial for the prevention and treatment of DCM. The upward arrow represents upregulation, and the downward arrow represents downregulation. The figure was created with BioRender.com, accessed on 30 January 2025.

**Figure 2 ijms-26-01465-f002:**
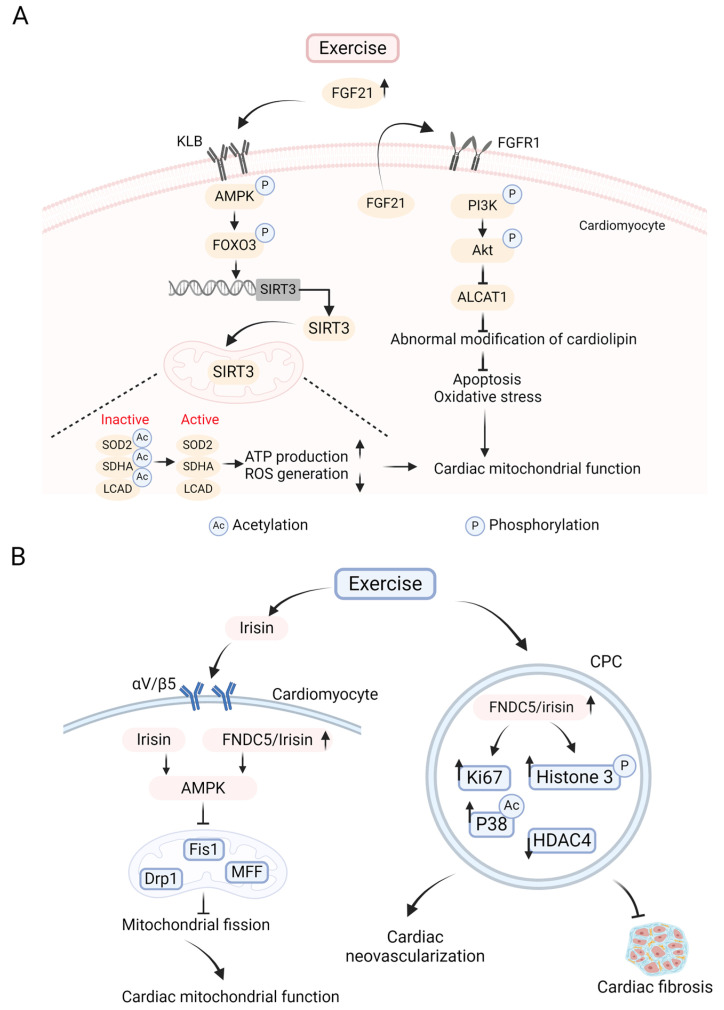
The protective role of fibroblast growth factor 21 (FGF21) and irisin in the exercise-mediated alleviation of DCM. (**A**) FGF21 is involved in the exercise-mediated improvement of DCM. Exercise upregulates the level of circulating FGF21 and enhances the sensitivity of its receptor KLB (β-klotho) on cardiomyocytes. After their combination, the AMPK signaling is activated to promote FOXO3 phosphorylation. FOXO3 directly transactivates the expression of SIRT3. The increased SIRT3 leads to the inhibition of the hyperacetylation of mitochondrial enzymes’ long-chain acyl-CoA dehydrogenase (LCDA), superoxide dismutase 2 (SOD2), and succinate dehydrogenase complex flavoprotein subunit A complex II subunits (SDHA) to enhance their activity. As a result, the production of ATP is enhanced, and the generation of ROS is reduced, which contributes to the improvement of cardiac mitochondrial function. On the other hand, in the heart with myocardial infarction (MI), exercise promotes cardiomyocytes to secrete FGF21. FGF21 binds to its receptor FGFR1 and then activates the phosphorylation of the PI3K-Akt pathway, which inhibits the expression of lysocardiolipin acyltransferase 1 (ALCAT1) to suppress the abnormal modification of cardiolipin, thereby reducing apoptosis and oxidative stress and enhancing the mitochondrial function of the hearts. However, in the diabetic heart, whether this mechanism is involved in the exercise-mediated improvement of DCM remains to be further investigated. (**B**) Exercise increases the level of serum irisin and myocardial FNDC5/irisin, which helps to alleviate DCM. Irisin binds to its myocardial receptor αV/β5 to activate the AMPK signal, which inhibits mitochondrial fission by suppressing the expression of dynamin-related protein 1 (Drp1), fission-1 protein (Fis1), as well as mitochondrial fission factor (MFF), thereby improving cardiac mitochondrial function. FNDC5/irisin in the myocardium can also activate the AMPK signaling and alleviate DCM through the above mechanisms. Additionally, exercise-induced myocardial FNDC5/irisin promotes the repair of damaged cardiomyocytes via Nkx2.5^+^ cardiac progenitor cells (CPCs), thus accelerating neovascularization and reducing cardiac fibrosis. This is related to the increase in Ki67, phosphorylated histone 3, as well as the acetylated p38, and the significant decrease in histone deacetylase 4 (HDAC4) in CPCs. Therefore, exercise-induced irisin may play a protective role against DCM by acting on CPCs. The upward arrow represents upregulation, and the downward arrow represents downregulation. The figure was created with BioRender.com, accessed on 30 January 2025.

**Figure 3 ijms-26-01465-f003:**
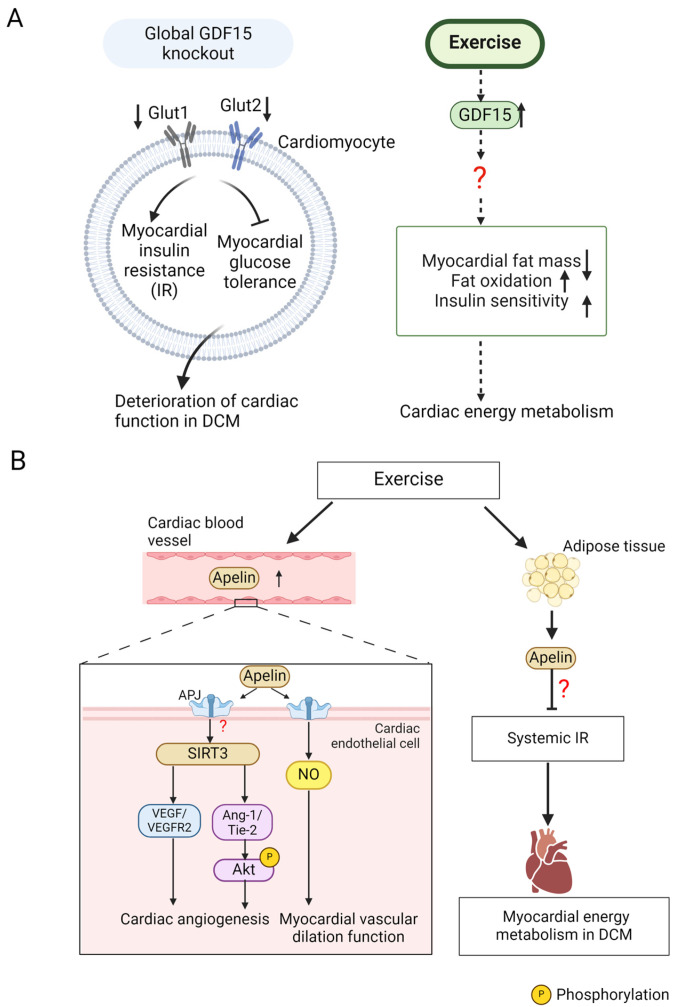
The role and mechanism of exercise-induced growth differentiation factor 15 (GDF15) and apelin in DCM. (**A**) In mice with global GDF15 knockout, the expressions of myocardial glucose transporters (GLUT1, GLUT2) are downregulated, which results in myocardial insulin resistance (IR) and reduced myocardial glucose tolerance, thereby inducing the DCM-related deterioration of cardiac function. Aerobic exercise raises the level of circulating GDF15. GDF15 reduces myocardial fat mass, promotes fat oxidation, and enhances insulin sensitivity in a certain way. This result indicates that exercise can induce an increase in the circulating level of GDF15 in elderly obese adults. However, whether exercise can increase the serum level of GDF15 in diabetic patients remains to be further investigated. (**B**) Exercise can increase the level of apelin to alleviate DCM. Apelin binds to the receptor APJ on cardiac endothelial cells, potentially activating the SIRT3 signal, which promotes the expression of vascular endothelial growth factor (VEGF) and its receptor VEGFR2. Apelin may also enhance the expression of Angiopoietin-1 (Ang-1) and its receptor Tyrosine kinase receptor with immunoglobulin-like and epidermal growth factor-like domain-2 (Tie-2) through SIRT3 activation, leading to the phosphorylation of Akt. These processes may collectively promote cardiac angiogenesis after exercise. Besides, the level of NO in the cardiac endothelial cells of heart tissues is increased after apelin binds to APJ, and exercise may improve the myocardial vascular dilation function by promoting this process to alleviate DCM. Additionally, exercise can increase the level of adipose tissue-derived apelin to reduce systemic IR, which may help to improve myocardial energy metabolism and ameliorate DCM. The upward arrow represents upregulation, and the downward arrow represents downregulation. The dashed arrow indicates that it remains to be confirmed in the DCM model. The red question mark means that the mechanism is not yet clear. The figure was created with BioRender.com, accessed on 30 January 2025.

**Figure 4 ijms-26-01465-f004:**
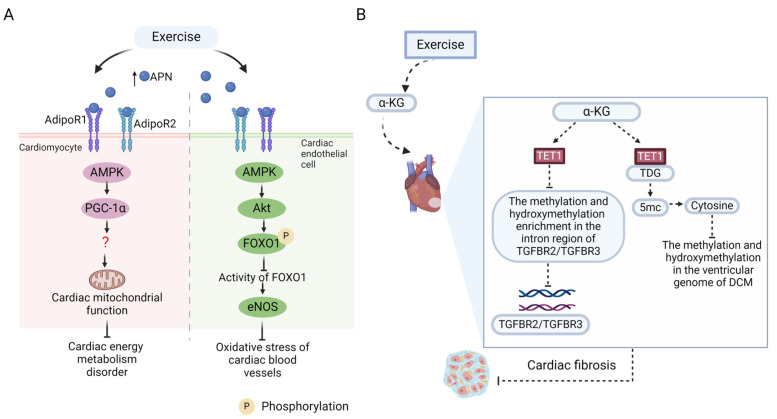
Exercise-induced adiponection (APN) and alpha-ketoglutarate (α-KG) signaling in DCM. (**A**) Exercise can elevate the serum level of APN, which may be beneficial for relieving DCM. In cardiomyocytes, APN may bind to its receptor adipoR1 and adipoR2 under exercise stimulation and then activates the AMPK signal to upregulate the expression of PGC-1α, which contributes to the improvement of mitochondrial function, thus reducing cardiac energy metabolism disorder induced by DCM. In cardiac endothelial cells, exercise may promote the binding of APN to adipoR1 and adipoR2, thereby activating the AMPK-Akt signal, which leads to the phosphorylation of FOXO1 (the downstream target of Akt) to inhibit the activity of FOXO1, thus promoting the production of eNOS to reduce the oxidative stress of cardiac blood vessels; this mechanism remains to be further verified. (**B**) α-KG has been proven to alleviate DCM as an exercise-induced metabolite. Mechanistically, α-KG promotes the binding of TET1 to the promoters of TGFBR2 and TGFBR3 genes, respectively, reducing the enrichment degrees of methylation and hydroxymethylation in the intron region of the profibrotic genes TGFBR2 and TGFBR3, thus downregulating their gene expressions and inhibiting cardiac fibrosis in DCM. In addition, TET1 also combines with TET to thymine DNA glycosylase (TDG) to form the TET1-TDG complex, which promotes the conversion of 5′ methylcytosine (5 mC) to cytosine, thereby alleviating the methylation and hydroxymethylation modifications of the genomic DNA in the left ventricular tissue of DCM. Exercise can upregulate the circulating level of α-KG. However, whether exercise can alleviate DCM through the above pathway remains to be confirmed. The upward arrow represents upregulation, and the dashed arrows indicate that it remains to be confirmed in the DCM model. The red question mark means that the mechanism is not yet clear. The figure was created with BioRender.com, accessed on 30 January 2025.

**Figure 5 ijms-26-01465-f005:**
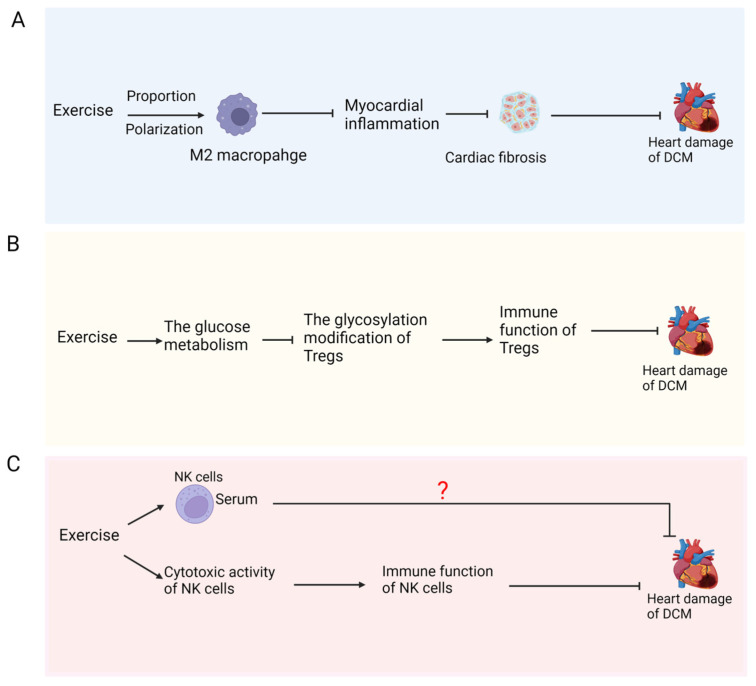
The involvement of immune-related factors in the amelioration of DCM by exercise. (**A**) Exercise may increase the proportion and polarization of M2 macrophages to reduce inflammation in diabetic hearts, thereby alleviating inflammation-induced fibrosis in DCM. (**B**) Exercise is involved in the amelioration of DCM via regulatory T cells (Tregs). Exercise improves the body’s glucose metabolism to reduce the glycosylation modification of Tregs, which enhances the immune function of Tregs and may alleviate cardiac inflammation in DCM. (**C**) Natural killer cells (NK cells) and their cytotoxic activity are involved in the exercise-mediated improvement of DCM. Exercise upregulates serum NK cells and may alleviate DCM in some way. In addition, exercise can enhance the cytotoxic activity of NK cells, thereby improving their immune function, which contributes to mitigating DCM. The red question mark indicate that the mechanism is not yet clear. The figure was created with BioRender.com, accessed on 30 January 2025.

**Figure 6 ijms-26-01465-f006:**
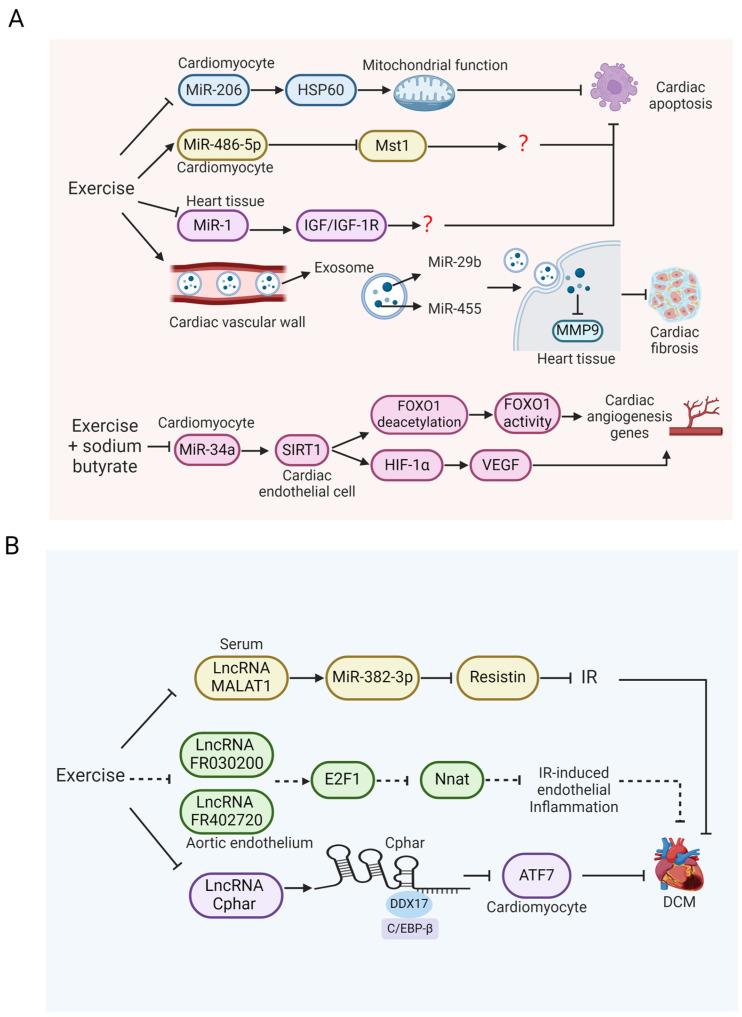
Cardioprotective effect of exercise-regulated microRNAs (miRNAs) and long non-coding RNAs (lncRNAs) on DCM. (**A**) MiRNAs play a significant role in exercise-mediated alleviation in DCM. Exercise inhibited the level of miR-206 in cardiomyocytes to increase the expression of heat shock protein 60 (HSP60), which reduces myocardial apoptosis. Exercise elevates the level of miR-486-5p to inhibit the expression of mammalian sterile 20-like kinase 1 (Mst1), which suppresses cardiac apoptosis. After exercise, the expression of miR-1 in cardiac tissues decreases, subsequently increasing the expression of insulin-like growth factor 1 (IGF) and its receptor IGF-1R, which helps alleviate cardiac apoptosis. Additionally, exercise facilitates the entry of more exosomes carrying miR-29b and miR-455 into the cardiac vascular wall. Subsequently, these exosomes enter the heart tissues and release miR-29b and miR-455 to inhibit the expression of matrix metalloprotease 9 (MMP9), thereby inhibiting cardiac fibrosis. Additionally, exercise combined with sodium butyrate treatment is able to suppress the expression of miR-34a in cardiomyocytes. Subsequently, this upregulates the expression of SIRT1 in cardiac endothelial cells to promote the deacetylation of FOXO1, thus enhancing the activity of FOXO1 and the expression of genes related to cardiac angiogenesis. On the other hand, SIRT1 promotes the expression of HIF-1α in endothelial cells and then increases the level of its downstream target VEGF to promote cardiac angiogenesis in DCM. (**B**) LncRNAs are important targets for exercise to relieve DCM. Exercise suppresses the serum level of lncRNA MALAT1, which increases miR-382-3p to decrease the level of resistin, thereby inhibiting systemic IR and exerting a cardioprotective effect in DCM. In the mouse model with IR induced by HFD, under exercise stimulation, the expression of lncRNA FR030200 and FR402720 in the aortic endothelium declines, which activates E2 promoter binding factor 1 (E2F1) to decrease the expression of neuronatin (Nnat), thereby alleviating IR-induced endothelial inflammation. Considering that the pathological process of diabetes is characterized by IR and vascular endothelial injury, it is hypothesized that exercise may decrease the risk of developing DCM by regulating some lncRNAs to improve IR, which deserves further investigation. Additionally, the expression of myocardial lncRNA Cphar is upregulated after exercise. LncRNA Cphar recruits DEAD-box helicase 17 (DDX17) to bind with C/EBP-β and then blocks the generation of activating transcription factor 7 (ATF7) in cardiomyocytes, leading to a decrease in the expression of ATF7, which may help to inhibit DCM. The red question mark indicates that the mechanism is not yet clear and the dashed arrows means that it remains to be confirmed in the DCM model. The figure was created with BioRender.com, accessed on 30 January 2025.

**Figure 7 ijms-26-01465-f007:**
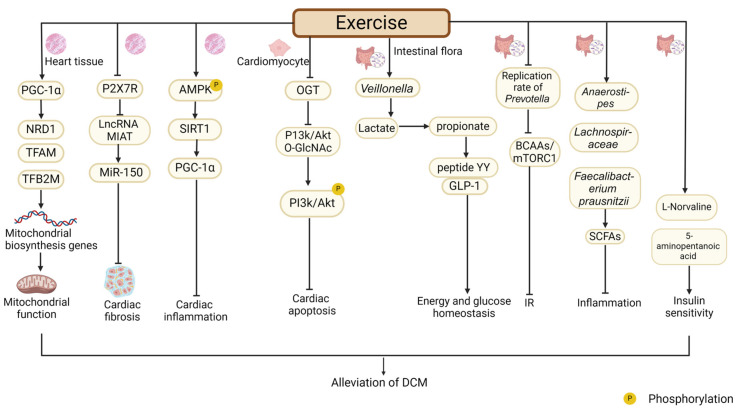
The role of other exercise-related effectors and processes in improving DCM. The level of PGC-1α in heart tissues is upregulated by exercise, which in turn promotes the expression of its downstream transcription factors NRD1, TFAM, and TFB2M, leading to an increase in the expression of genes related to mitochondrial biosynthesis in DCM. Exercise inhibits the expression of P2X7R in diabetic hearts and then suppresses the level of lncRNA MIAT to upregulate the expression of miR-50, which alleviates cardiac fibrosis in DCM. Exercise may alleviate DCM by upregulating the expression of SIRT1. In heart tissues with MI in T2DM, exercise can activate the phosphorylation signal of AMPK and then upregulate the expression of SIRT1 to promote the production of PGC-1α, thereby reducing inflammation. Moreover, exercise may reduce the cardiac expression of O-GlcNAc transferase (OGT), which weakens the O-GlcNAcylation of PI3k and Akt to promote their phosphorylation and then increases their activity, thus suppressing cardiac dysfunction in DCM. Additionally, after exercise, the number of *Veillonella* increased, decomposing lactate into propionate in the intestine. Propionate induces the secretion of peptide YY and glucagon-like peptide-1 (GLP-1), which are essential for maintaining energy and glucose homeostasis in DCM. The exercise-induced low replication rate of *Prevotella* weakens the biosynthesis of branched chain amino acids (BCAAs) and inhibits BCAAs/mTORC1-mediated IR, potentially preventing DCM. Exercise can also significantly enrich *Anaerostipes*, *Lachnospiraceae*, and *Faecalibacterium prausnitzii* to produce short-chain fatty acids (SCFA) to ameliorate inflammation. Besides, the level of 5-aminopentanoic acid and L-Norvaline are elevated after exercise, playing beneficial roles in promoting insulin sensitivity and controlling blood glucose. Thus, exercise may prevent and treat DCM by regulating the composition and function of intestinal flora. The figure was created with BioRender.com, accessed on 30 January 2025.

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
