# Peer review of "Exercise in Diabetic Cardiomyopathy: Its Protective Effects and Molecular Mechanism"

_ijms, 2025, doi:10.3390/ijms26041465_

Round 1

Reviewer 1 Report

Comments and Suggestions for Authors

This is a review aiming to summarize the current knowledge about the molecular protective mechanisms of exercises in diabetic cardiomyopathy. The advantage of this paper is the aim of summarize a lot of important information. Authors included 171 references in total, however not all really related to the topic. Authors mixed the information about diabetes, metabolic syndrome, obesity, myocardial infarction, and even high-fat feeding. I would recommend to focus on diabetes and also to differentiates between diabetes type 1 and type 2. Authors uploaded 7 beautiful figures, however they remain somehow apart from the main text. Authors have to incorporate the figures into the text and try to organize the literature findings around those figures.

Comments:

1. p.2. - Ref 19 is wrongly cited as meta-analysis. 

2. p.2. - Ref 20 is about 7% of patients with diabetes and not 15%. 

3. Reff 21 and 21 are NOT related to the miRNA-24. The results from those two papers are not cited properly.

4. p. 2 - "In summary, mounting evidence supports that exercise alleviates cardiac dysfunction in patients with DCM. The frequency, intensity, and type of exercise have been shown to be important influencing factors."- references are missing here. 

5. p.3. Ref 42 is about high-fat fed rats but NOT for diabetes.

6. p. "4 4.1. The contribution of exercise-induced hormones and metabolites..." . Actually, hormones are not discussed in this subsection. Cytokines are discussed.

7.  From Line 193 to Line 269 no one Reference exist. How it could be?

8. p. 7. Ref 61 is wrongly cited here. No patients with diabetes in the cited paper. Their glucose is 5.67 mmol/l. Metabolic syndrome is not equal of diabetes. 

9. Ref. 63 is about human beans, however it was placed as a reference for rodent experiment.

10. Ref. 66 is about obese subjects and NOT subject with diabetes.

11. Ref. 70 is about patients with diabetes, however they are compared with obese mice (Ref. 71). Obesity could be a confounding factor. How many patients from Ref. 71 were with obesity + diabetes?

12. Ref. 87 is about obese mice fed with HFD. Obesity is NOT equal of diabetes.

13. Ref. 91 is about adipose tissue in obese mice. No diabetes, no heart.

14. Ref. 98 is about striated muscles performance rather than the myocardium. 

15. Ref. 104 is about NK cells itself. Not about diabetes, neither about heart.

16. Ref. 106 is about doxorubicin. This is also out of the scope here.

17. p.11 - Ref. 108 - not related with diabetic cardiomyopathy. The paper is about potential inhibitors of HSP60 with anti-cancer, anti-inflammatory and anti-autoimmune properties. "Exercise inhibited the expression of miR-206 in left ventricular tissues, and then elevated HSP60 protein expression to suppress the apoptosis of ventricular myocytes." - please, add Ref.

18. p. 11 - " However, 12-week treadmill exercise significantly upregulated the expression of miR-486-5p in heart tissues, with the reduction of left ventricular end-diastolic diameter (LVEDD) and left ventricular end-systolic diameter (LVESD), the increase of EF and the remission of DCM cardiac dysfunction" - please, add Ref.

19. p.12 - "Silva et al [118]." Ref. 118 is from Sarlak et al.

20. p. 12 Ref. 125 is for type 2 diabetes model, NOT type 1. (Ref. 125).

21. p.12. -"Swimming has been proved to reverse this phenomenon"- please, add Ref.

22. Ref. 128 is about high-fat diet, NOT for diabetes. Insulin resistance is not equal of diabetes. 

23. p.13 - "4.5. The role of other exercise-related effectors and processes in improving DCM"- under this subheading  are added PGC-1alfa, however its link with physical activity was not supported by research papers.

24. Ref. 141 is about myocardial infarction rather that diabetic myocardiopathy.  No diabetic mice in the cited paper. 

25. p. 14 - "Compared to the DCM group, mice in the DCM+P2X7R KO group showed lower lncRNA MIAT expression and higher miR-150 expression. Thus, it is speculated that the P2X7R/MIAT/miR-150 pathway may be a downstream mechanism through which exercise ameliorates DCM."- please, add the Ref.

26. p.14 - the information about doxorubicin is irrelevant to the topic of this manuscript. Doxorubicin is a well-known cardiotoxic drug. Please, add studies about diabetic cardiopathy, physical activity and ferroptosis.

27. p.15 - "Shiva et al [155]"in the bibliography is written as Roshan Milani et al. Please, correct.

28. Ref. 158 is about astrocytes. Out of the scope of this manuscript. 

29. p.15 - "It has been found that the level of O-GlcNAcylation of total proteins in the myocardium of diabetic patients was higher than that of non-diabetic patients and was closely connected with left ventricular dysfunction"- please, add Ref.

Author Response

General Question :

However not all really related to the topic. Authors mixed the information about diabetes, metabolic syndrome, obesity, myocardial infarction, and even high-fat feeding. I would recommend to focus on diabetes and also to differentiates between diabetes type 1 and type 2. Authors uploaded 7 beautiful figures, however they remain somehow apart from the main text. Authors have to incorporate the figures into the text and try to organize the literature findings around those figures.

Answer:

Thank you for your careful reading and valuable suggestions for the paper. After careful review, we have removed the weakly-relevant examples and citations to stay on topic. Also, we have clearly marked whether the cited examples were about T1DM or T2DM. Finally, we have made some modifications to the figures to ensure that they are not apart from the main text, which are shown as following.

Fig.3A

As shown on the right side of Fig3A, exercise can upregulate the level of serum GDF15 in obese old adults and brings about a series of benefits which helps to decrease the risk of DCM. However, this has not been verified in the diabetic heart yet, so it is represented by a dotted line instead.

Fig.4B

As shown in the figure, the molecular mechanism by which α-KG alleviates diabetic cardiac fibrosis has been verified in mice with DCM. Exercise has been verified to elevate the level of serum α-KG in obese mice fed with HFD. Obese is a high-risk factor for diabetes. However, whether exercise can exert a protective effect on DCM through this pathway remains to be explored. Therefore, it is represented by a dotted line.

Fig.5B

The reference (originally Ref.98) for the statement in the original manuscript, "Exercise may alleviate DCM by increasing the potential of initial CD4+ T cells to differentiate into Tregs" pertains to striated muscles performance rather than the myocardium. It has little relevance to the theme. Therefore, this content was removed from the figure.

Fig.5C

In the original manuscript, the reference (originally Ref.104) for the content "An exercise experiment was conducted on 60 healthy sedentary men, and it was found that exercise elevated the expression of cytotoxic granule proteins and the level of mitochondrial membrane potential in NK cells" is about NK cell itself and has no relation to diabetes and DCM. Therefore, we removed this content from the figure.

Fig.6B

Whether the LncRNA FR030200/FR402720-E2F1-Nnat pathway can exert protective effects in the diabetic heart under exercise stimulation remains to be confirmed. However, they have been verified in the hearts of mice with IR induced by HFD, and IR is the important pathological process of diabetes. Therefore, this content is represented by dotted lines instead.

Fig7.

In the original manuscript, the content "4.5.3 Exercise-mediated inhibition of ferroptosis in DCM" mainly referred to literature on doxorubicin, which has little relevance to DCM. Therefore, we removed this content from both the text and the figure.

Question #1:

p.2. - Ref 19 is wrongly cited as meta-analysis. 

Answer:

We're really sorry for this mistake. Thank you for your careful reading. In the original manuscript, Ref.19 was a wrong citation. In the revised manuscript, it has been corrected to the correct citation (Ref. 19), which is a meta-analysis paper.

Question #2:

p.2. - Ref 20 is about 7% of patients with diabetes and not 15%. 

Answer:

We're really sorry for this mistake. Thank you for your careful reading. In the revised manuscript, we have corrected “15%” to “7%”.

Question #3:

Ref. 21 and 21 are NOT related to the miRNA-24. The results from those two papers are not cited properly.

Answer:

We have checked Ref.21 and 22. Indeed, they are not closely related to the theme. Besides, we couldn't find any other relevant literature. Therefore, the content about miR-24 and Ref.20 as well as Ref.21 has been deleted.

Question #4:

p.2-"In summary, mounting evidence supports that exercise alleviates cardiac dysfunction in patients with DCM. The frequency, intensity, and type of exercise have been shown to be important influencing factors."- references are missing here. 

Answer:

We're really sorry for this mistake. We have added appropriate citations (Ref. 25,28,29,19,20) at the end of these two sentences respectively according to your suggestions.

Question #5:

p.3. Ref 42 is about high-fat fed rats but NOT for diabetes.

Answer:

Thank you for your careful reading. We have verified that the research subjects in Ref.42 are indeed rats fed with a HFD, not diabetic ones. Therefore, we have deleted the relevant content and the citation.

Question #6:

p."4 4.1. The contribution of exercise-induced hormones and metabolites...". Actually, hormones are not discussed in this subsection. Cytokines are discussed.

Answer:

Thank you for your careful reading. We have changed “hormones” to “cytokines” according to your suggestion.

Question #7:

From Line 193 to Line 269 no one Reference exist. How it could be?

Answer:

We apologize for overlooking this. Citations were missing from line 193 to line 269. We have now added the corresponding citations (Ref.29,35,41,51,45,47,30,32,54-63) to the relevant content.

Question #8:

  1. 7. Ref 61 is wrongly cited here. No patients with diabetes in the cited paper. Their glucose is 5.67 mmol/l. Metabolic syndrome is not equal of diabetes. 

Answer:

We have verified Ref.61 and found that it has little relevance to diabetes. Therefore, we have deleted this part of the content and the citation.

Question #9:

Ref. 63 is about human beans, however it was placed as a reference for rodent experiment.

Answer:

We have verified Ref.63 (Now it is Ref.74) and confirmed that it is about humans. Accordingly, we have changed “mice” in the original text to “asymptomatic patients”.

Question #10:

Ref. 66 is about obese subjects and NOT subject with diabetes.

Answer:

We have verified that Ref.66 (Now it is Ref.76) is indeed about obese subjects. However, here we aim to demonstrate that exercise can promote an increase in the circulating GDF15 level. Since we couldn't find direct evidence that exercise upregulates the serum GDF15 level in people with the diabetic state, and considering that obesity is a high-risk factor for diabetes and DCM, we have revised it to "This result indicates that exercise can induce an increase in the circulating level of GDF15 in elderly obese adults. However, whether exercise can increase the serum level of GDF15 in diabetic patients remains to be further investigated." We hope that the reviewer could agree.

Question #11:

Ref. 70 is about patients with diabetes, however they are compared with obese mice (Ref. 71). Obesity could be a confounding factor. How many patients from Ref. 71 were with obesity + diabetes?

Answer:

We agree with the reviewer that patients with diabetes can’t be compared with obese mice. Furthermore, obesity is not equal to diabetes and Ref.71 is about mice studies and did not mention patients with both obesity and diabetes. Therefore, the content related to Ref.71 and Ref.71 have been deleted. We hope that the reviewer could agree.

Question #12:

Ref. 87 is about obese mice fed with HFD. Obesity is NOT equal of diabetes.

Answer:

The research subjects in Ref.87 (Now it is Ref.97) are indeed obese mice fed with a HFD. Here, the aim was to prove that "exercise can upregulate α-KG". Since there is no direct evidence found for exercise to upregulate α-KG in mice or patients with diabetes and DCM, and given that obesity is a high-risk factor for diabetes, the original text here has been revised to "Exercise can upregulate the serum level of α-KG. However, it remains unknown whether such changes in α-KG occur under diabetic conditions. It is speculated that exercise may alleviate diabetes by elevating α-KG, thereby reducing the risk of DCM, which needs to be proved in the future studies."

Question #13:

Ref. 91 is about adipose tissue in obese mice. No diabetes, no heart.

Answer:

We've verified that Ref.91 has little connection with diabetes and DCM. Now, both Ref. 91 and its relevant content have been deleted.

Question #14:

Ref. 98 is about striated muscles performance rather than the myocardium. 

Answer:

Thank you for your careful reading. It has been verified that Ref.98 is about striated muscles rather than myocardium. Therefore, Ref.98 and its related content have been deleted.

Question #15:

Ref. 104 is about NK cells itself. Not about diabetes, neither about heart.

Answer:

Thank you for your careful reading. Ref.104 and its related content have been deleted according to your suggestion.

Question #16:

Ref. 106 is about doxorubicin. This is also out of the scope here.

Answer:

Thank you for your careful reading. In order to stay on topic, we have deleted Ref.106.

Question #17:

p.11-Ref.108-not related with diabetic cardiomyopathy. The paper is about potential inhibitors of HSP60 with anti-cancer, anti-inflammatory and anti-autoimmune properties. "Exercise inhibited the expression of miR-206 in left ventricular tissues, and then elevated HSP60 protein expression to suppress the apoptosis of ventricular myocytes." - please, add Ref.

Answer:

We have verified that Ref.108 is not relevant to DCM, so we have deleted it. Thank you for your careful reading. A citation (Ref.117) has been added at the end of the sentence "Exercise inhibited the expression of miR-206 in left ventricular tissues, and then elevated HSP60 protein expression to suppress the apoptosis of ventricular myocytes."

Question #18:

p.11-" However, 12-week treadmill exercise significantly upregulated the expression of miR-486-5p in heart tissues, with the reduction of left ventricular end-diastolic diameter (LVEDD) and left ventricular end-systolic diameter (LVESD), the increase of EF and the remission of DCM cardiac dysfunction" - please, add Ref.

Answer:

Thank you for your careful reading. We have added an appropriate citation (Ref.120) at the end of this sentence according to your suggestion.

Question #19:

p.12 - "Silva et al [118]." Ref. 118 is from Sarlak et al.

Answer:

Thank you for your careful reading. We apologize for mistaking this. "Silva" has been changed to "Sarlak" (Now it is Ref.127).

Question #20:

  1. 12 Ref. 125 is for type 2 diabetes model, NOT type 1. (Ref. 125).

Answer:

Thank you for your careful reading. We apologize for mistaking this. The original "T1DM" in the text has been changed to "T2DM" (Now it is Ref.134).

Question #21:

p.12. -"Swimming has been proved to reverse this phenomenon"- please, add Ref.

Answer:

Thank you for your careful reading. A citation (Ref. 134) has been added at the end of the sentence "Swimming has been proved to reverse this phenomenon."

Question #22:

Ref. 128 is about high-fat diet, NOT for diabetes. Insulin resistance is not equal of diabetes.

Answer:

We have verified Ref. 128 (Now it is Ref.137), which is indeed about insulin-resistant (IR) mice fed with a HFD. Therefore, the sentence "thereby reducing the risk of DCM associated complications" has been revised to "thereby reducing the risk of IR-related cardiac injury." Moreover, at the end of this paragraph, the sentence “Considering that the pathological process of diabetes is characterized by IR and vascular endothelial injury, it is hypothesized that exercise may decrease the risk of developing DCM by regulating some lncRNAs to improve IR, which deserves further investigation” has been included. We hope that the reviewer could agree.

Question #23:

p.13 - "4.5. The role of other exercise-related effectors and processes in improving DCM"- under this subheading are added PGC-1alfa, however its link with physical activity was not supported by research papers.

Answer:

We have found research papers demonstrating that exercise can upregulate PGC-1α in the diabetic heart, thus alleviating DCM. An example is the paper "Exercise enhances cardiac function by improving mitochondrial dysfunction and maintaining energy homoeostasis in the development of diabetic cardiomyopathy"(doi: 10.1007/s00109-019-01861-2). We have added relevant content from this literature in the PGC-1α section, which is shown as following “Moreover, a study reported that in mice with T2DM-induced DCM, exercise promoted the expression of PGC-1α in cardiomyocytes to improve mitochondrial dysfunction and energy metabolism, which enhanced cardiac function146”. We hope that the reviewer could agree.

Question #24:

Ref. 141 is about myocardial infarction rather that diabetic myocardiopathy.  No diabetic mice in the cited paper. 

Answer:

Thank you for your careful reading. Ref.141 and its related content have been deleted according to your suggestion.

Question #25:

  1. 14 - "Compared to the DCM group, mice in the DCM+P2X7R KO group showed lower lncRNA MIAT expression and higher miR-150 expression. Thus, it is speculated that the P2X7R/MIAT/miR-150 pathway may be a downstream mechanism through which exercise ameliorates DCM."- please, add the Ref.

Answer:

Thank you for your careful reading. We apologize for mistaking this. We have added an appropriate citation (Ref.152) at the end of this sentence.

Question #26:

p.14 - the information about doxorubicin is irrelevant to the topic of this manuscript. Doxorubicin is a well-known cardiotoxic drug. Please, add studies about diabetic cardiopathy, physical activity and ferroptosis.

Answer:

The content on ferroptosis was explained solely through doxorubicin-related literature, which did deviate from DCM. Since no more suitable references could be found, all of the content regarding ferroptosis has been deleted.

Question #27:

p.15 - "Shiva et al [155]"in the bibliography is written as Roshan Milani et al. Please, correct.

Answer:

Thank you for your careful reading. We apologize for mistaking this. “Shiva” has been changed to “Roshan Milani” (Now it is Ref.170).

Question #28:

Ref. 158 is about astrocytes. Out of the scope of this manuscript. 

Answer:

Thank you for your careful reading. Ref.158 has been deleted.

Question #29:

p.15 - "It has been found that the level of O-GlcNAcylation of total proteins in the myocardium of diabetic patients was higher than that of non-diabetic patients and was closely connected with left ventricular dysfunction"- please, add Ref.

Answer:

Thank you for your careful reading. We apologize for mistaking this. We have added an appropriate citation (Ref.173) at the end of this sentence.

Reviewer 2 Report

Comments and Suggestions for Authors

The manuscript presents an interesting and well-written review on the role of exercise as a sustainable therapeutic approach for DCM. The authors provide a comprehensive overview of the pathophysiology of DCM, emphasizing key contributing factors such as inflammation, oxidative stress, fibrosis, and impaired glucose metabolism. The discussion on the beneficial effects of exercise in regulating blood glucose levels and improving energy metabolism through various molecular mechanisms is particularly insightful and well-supported by recent literature.

However, I would suggest adding further details on the pathophysiology of left ventricular remodeling, particularly focusing on the mechanisms that lead to the echocardiographic alterations detectable via TDI. A more in-depth discussion of the structural and functional changes could enhance the reader’s understanding of how DCM progresses and how it can be monitored clinically. Additionally, it would be beneficial to explore the analogies between the diabetic heart and the aging heart (senile heart), as both conditions share common pathophysiological features such as myocardial stiffening, fibrosis, and diastolic dysfunction. Highlighting these similarities could provide a broader perspective on the impact of metabolic and age-related changes on cardiac function.

Overall, the manuscript is well-organized, concise, and informative, making a significant contribution to the current understanding of exercise interventions in DCM. 

Author Response

Question #1:

However, I would suggest adding further details on the pathophysiology of left ventricular remodeling, particularly focusing on the mechanisms that lead to the echocardiographic alterations detectable via TDI. A more in-depth discussion of the structural and functional changes could enhance the reader’s understanding of how DCM progresses and how it can be monitored clinically. Additionally, it would be beneficial to explore the analogies between the diabetic heart and the aging heart (senile heart), as both conditions share common pathophysiological features such as myocardial stiffening, fibrosis, and diastolic dysfunction. Highlighting these similarities could provide a broader perspective on the impact of metabolic and age-related changes on cardiac function.

Answer:

We appreciate the reviewer’s insightful comments. We have added a "discussion" section (5. Discussion), which is divided into two parts: " 5.1 The pathophysiology of cardiac remodeling in DCM that leads to the echocardiographic alterations detectable via Tissue Doppler Imaging (TDI)" and "5.2 The similarities between the diabetic heart and the aging heart". We hope this part can help readers better understand how DCM progresses and how it can be monitored clinically, as well as the connection between the impacts of metabolism and age-related changes on cardiac function.

If you have any suggestions on the content of the discussion, like adding more details or adjusting the structure, feel free to let me know.

Round 2

Reviewer 1 Report

Comments and Suggestions for Authors

Most of the Reviewer's suggestions were implemented. Thank you for this.

Minor revision:

1. Ref. 19 is NOT meta-analysis. It is a systematic review - p.2 L.59. Please, correct.

2. All figures have to be placed into the manuscript next to the corresponding text, rather than at the end of the manuscript. Please, do it. 

Author Response

Comment #1:

  1. 19 is NOT meta-analysis. It is a systematic review - p.2 L.59. Please, correct.

Answer:

Thank you for your careful reading. We're sorry for this mistake. We have corrected "Ref.19 meta-analysis" to "systematic review", and the correction has been highlighted in yellow in the text.

Comment #2:

  1. All figures have to be placed into the manuscript next to the corresponding text, rather than at the end of the manuscript. Please, do it. 

Answer:

Thank you for your careful reading. We're sorry for this mistake. We have placed the figures and figure legends at the corresponding positions in the manuscript. Please check.